# IMC-Denoise: a content aware denoising pipeline to enhance Imaging Mass Cytometry

Peng Lu [1,2,3,11], Karolyn A. Oetjen [4,11], Diane E. Bender [5], Marianna B. Ruzinova[6], Daniel A. C. Fisher[4], Kevin G. Shim[4], Russell K. Pachynski[4], W. Nathaniel Brennen [7,8], Stephen T. Oh [4,5,6], Daniel C. Link [4,6] & Daniel L. J. Thorek [1,2,3,9,10] ✉

Imaging Mass Cytometry (IMC) is an emerging multiplexed imaging technology for analyzing complex microenvironments using more than 40 molecularly-specific channels. However, this modality has unique data processing requirements, particularly for patient tissue specimens where signal-to-noise ratios for markers can be low, despite optimization, and pixel intensity artifacts can deteriorate image quality and downstream analysis. Here we demonstrate an automated content-aware pipeline, IMC-Denoise, to restore IMC images deploying a differential intensity map-based restoration (DIMR) algorithm for removing hot pixels and a self-supervised deep learning algorithm for shot noise image filtering (DeepSNiF). IMC-Denoise outperforms existing methods for adaptive hot pixel and background noise removal, with significant image quality improvement in modeled data and datasets from multiple pathologies. This includes in technically challenging human bone marrow; we achieve noise level reduction of 87% for a 5.6-fold higher contrast-to-noise ratio, and more accurate background noise removal with approximately 2 × improved F1 score. Our approach enhances manual gating and automated phenotyping with cell-scale downstream analyses. Verified by manual annotations, spatial and density analysis for targeted cell groups reveal subtle but significant differences of cell populations in diseased bone marrow. We anticipate that IMC-Denoise will provide similar benefits across mass cytometric applications to more deeply characterize complex tissue microenvironments.

Disease states are the result of a complex interplay of many different cell types interacting in close proximity in the context of often heterogeneous tissues. Traditional approaches to study these features at the tissue scale have been limited in the number of specific markers that can be acquired to robustly resolve distinct cell types. Flow cytometry, perhaps the most widely used technique to study cell populations and states in this milieu, requires single-cell disaggregation of the tissue resulting in complete loss of spatial context[1,2]. Highly multiplexed imaging provides a means to assess these events at cellular resolution in situ, with extensive protocol development in progress[3], including tissue-based cyclic immunofluorescence (t-CyCIF)[4], co-detection by indexing (CODEX)[5], Multiplexed Ion Beam Imaging (MIBI)[6,7], and Imaging Mass Cytometry (IMC)[8]. In IMC, tissue sections are stained with a panel of metal-conjugated antibodies, and data is acquired by UV-laser raster ablation of the section in 1-micron pixels for cytometry by time-of-flight (CyTOF) mass analyzer. This novel imaging technology allows for the detection of more than 40 antigens simultaneously to facilitate single-cell, spatially resolved,

highly multiplexed analysis of solid tissues. This provides essential information on the distribution of transcripts, proteins, and protein modifications within single cells, microenvironments, and entire tissues[8–17]. The pixel data is processed into an image, thereby allowing the visualization of phenotypes and incorporation of spatial information in subsequent analyses. These properties make it a unique tool for the evaluation of complex biological systems.

Despite the wide applications in pre- and clinical research using this state-of-the-art multiplexed imaging technique, there exist specific technical noise sources in IMC, which include hot pixels, channel spillover and shot noise[8–10,15,18,19]. Hot pixels are concentrated areas of high counts which are uncorrelated with any biological structures. Putatively, these can result from deposition of metal-stained antibody aggregates. In IMC images, single hot pixels are the most common outliers, and small hot clusters with several consecutive pixels may also exist. Channel spillover refers to scenarios where the signal of a source channel contaminates a target channel or is correlated with such contamination. The spillover in IMC can occur from a variety of reasons, such as instrument properties (abundance sensitivity), isotopic impurities and oxidation. Finally, shot noise exists because of ion counting imaging processes, which are pixel-independent, signal-dependent and usually modeled as a Poisson process. Additionally, noise levels are related to multiple other factors, including variations in conjugated metal isotopes, antibody concentration and arrangement.

Together these noise sources appreciably deteriorate image quality and distort downstream analyses of IMC data. Differing from traditional fluorescence-based imaging modalities, there are low background features and no read-out noises from imaging sensors in IMC. A number of studies have attempted to address the unique imaging data features of IMC. Hot pixels can be corrected by thresholding methods[10,14,15,20]; however, due to the differences between marker channels and tissues, a threshold needs to be pre-set carefully. An inappropriate threshold may lead to unsatisfactory results. Post-acquisition methods[10,19] and a bead-based compensation workflow[18] have been proposed to correct the channel spillover phenomenon. However, spillover correction may not be necessary if the marker panel employed is well-designed and titrated; and the intensity of channel-overlapping signal is often weak[18]. Therefore, spillover can be neglected when using low concentrations of staining antibodies, which however further lowers signal-to-noise ratio (SNR). To account for the impact of shot noise, MAUI[7,19] and a semi-automated Ilastik-based method[21] have been used for background noise removal. These approaches require finely tuned parameters or manually annotated background regions, requiring preprocessing expertize. In tissues with low marker signals, highly intermixed cell populations, or difficult immunostaining defining thresholds can be time consuming with high inter-user subjectivity, which may still result in poor image quality that complicates further analyses.

In the present work we develop and apply IMC-Denoise, a content aware denoising pipeline to enhance IMC images through an automated process. To account for the two major noise sources in this modality, hot pixels and shot noise, IMC-Denoise invokes novel algorithms for differential intensity map-based restoration (DIMR) and self-supervised deep learning-based shot noise image filtering (DeepSNiF). We demonstrate the flexibility and effectiveness of the proposed pipeline on publicly available IMC datasets of pancreatic cancer[10], breast cancer[12], a MIBI dataset[19], and deploy it on a technically challenging unique human bone marrow dataset. We benchmark our approach against existing hot pixel removal methods[10,14,15,20] and other advanced biomedical imaging denoising algorithms, such as non-local means filtering (NLM)[22], batch matching and 3D filtering (BM3D)[23] and Noise2Void (N2V)[24], which is used in IMC here for the first time. We demonstrate that the image formation model derived IMC-Denoise pipeline produces image quality enhancements that are best-in-class and leads to improved downstream analysis, with limited manual user

manipulation. Qualitative improvements in images enhances their interpretation, and quantitatively improve molecularly-defined phenotyping. Results from the IMC-Denoise pipeline are suitable for further downstream analysis, such as Mesmer/DeepCell and ark-analysis[25] or MCMicro[26]. The IMC-Denoise software package and the corresponding tutorial have been published on Github (https://github.com/PENGLU-WashU/IMC_Denoise). We provide this tool to augment studies that seek to more deeply characterize the complex and diverse tissue microenvironment.

## Results

### IMC-Denoise principle

The general principle of IMC-Denoise is schematized in Fig. 1a and Supplementary Notes 1. To account for hot pixels and shot noise, an accurate IMC imaging joint model is built as Eq. (1), by considering ion counting imaging as a Poisson process (Supplementary Notes 1.1).

$$\mathbf{R} = \mathcal{P}[\mathbf{X} + \mathbf{X}^{\text{spillover}}] + \mathbf{Q}, \tag{1}$$

where $\mathbf{R}$ is the raw image, $\mathbf{X}$ the "clean" signal, $\mathbf{X}^{\text{spillover}}$ the spillover signals without noise, $\mathcal{P}[x]$ the Poisson noise with mean $x$, and $\mathbf{Q}$ the hot pixels. The term $\mathbf{X}^{\text{spillover}}$ in Eq. (1) can be omitted if the spillover is limited, which is often the case. However, the magnitude of image degradation from hot pixel and shot noise sources is considerable, resulting in bias and errors in downstream analysis and addressed in turn, below (Supplementary Notes 1.2).

In IMC-Denoise, the DIMR algorithm (Fig. 1ai and Supplementary Notes 1.3) builds differential maps to detect the hot pixels by comparing adjacent pixels in a 3 × 3 sliding window, as hot pixels are local maxima. The Anscombe transformation[27] is applied to the raw image $\mathbf{R}$ followed by background removal of intensities lower than 4 (for IMC), so that the difference between adjacent pixels, $\mathbf{D}_i$, can be feasibly approximated as a generalized Gaussian distribution[28], where $i$ is the neighbor index in the sliding window ($i \in \{1, 2, ..., 8\}$). Additionally, as with all biomedical imaging acquisition, in IMC datasets the tissue or background pixels should be continuous. Under these conditions, for a specific pixel $p$ there must exist several $d_i^p$ close to the mean $\mu_i$ of its corresponding distribution $\mathbf{D}_i$, except in the presence of a hot pixel. To unmix outliers from normal pixels, we consequently calculate the distances between $d_i^p$ and $\mu_i$ as $\triangle_i^p = |d_i^p - \mu_i|$ and sort $\triangle_i^p$ for $i \in \{1, 2, ..., 8\}$. Then, the $d_i^p$ corresponding to the first $l$ smallest $\triangle_i^p$ are summed, and the results from all pixels form a new distribution, $\mathbf{T}_l$. Compared to those in the distributions $\mathbf{D}_i$, the hot pixels move beyond the right tail of $\mathbf{T}_l$, while the normal relevant pixels move towards its center (Supplementary Note 1.3.1). To robustly detect the outliers, the kernel density estimation algorithm[29] is applied to $\mathbf{T}_l$ afterwards (Supplementary Note 1.3.2). On the fitted curve $(x, \hat{g}_h(x))$, a threshold point $x_T$ is defined so that any points $x > x_T$ are considered as outliers and filtered by a 3 × 3 median filter. Because outliers are located beyond the right tail of $\mathbf{T}_l$, it is reasonable to set $x_T$ when $\frac{d\hat{g}_h(x)}{dx} \to 0$, which means the current distribution ends. Likewise, the shape of the distribution should not change from convex to concave on the right tail. Thus, it is also reasonable to set $x_T$ when $\frac{d^2\hat{g}_h(x-\triangle x)}{dx^2} \geq 0$ and $\frac{d^2\hat{g}_h(x)}{dx^2} \leq 0$, where $\triangle x$ represents a small value. Because the pixel values of the raw images are discrete, $\triangle x$ is normally set as 1. We operate DIMR for multiple iterations to adequately remove hot pixels until no outliers are detected. The hot pixel removed images are transformed to their original scales with the direct algebraic inverse Anscombe transformation[30]. The DIMR algoirthm is summarized as Supplementary Algorithm 1. In the implementation, we use the median $\tilde{\mu}_i$ of distribution $\mathbf{D}_i$ as a robust estimation of the mean $\mu_i$. In addition, it is normally assumed that at least half of the neighbors are close to the center pixel in a 3 × 3 window so that $l = 4$[31,32]. Validated by simulation (Supplementary Note 3.3.1), the iteration number is set as 3 to

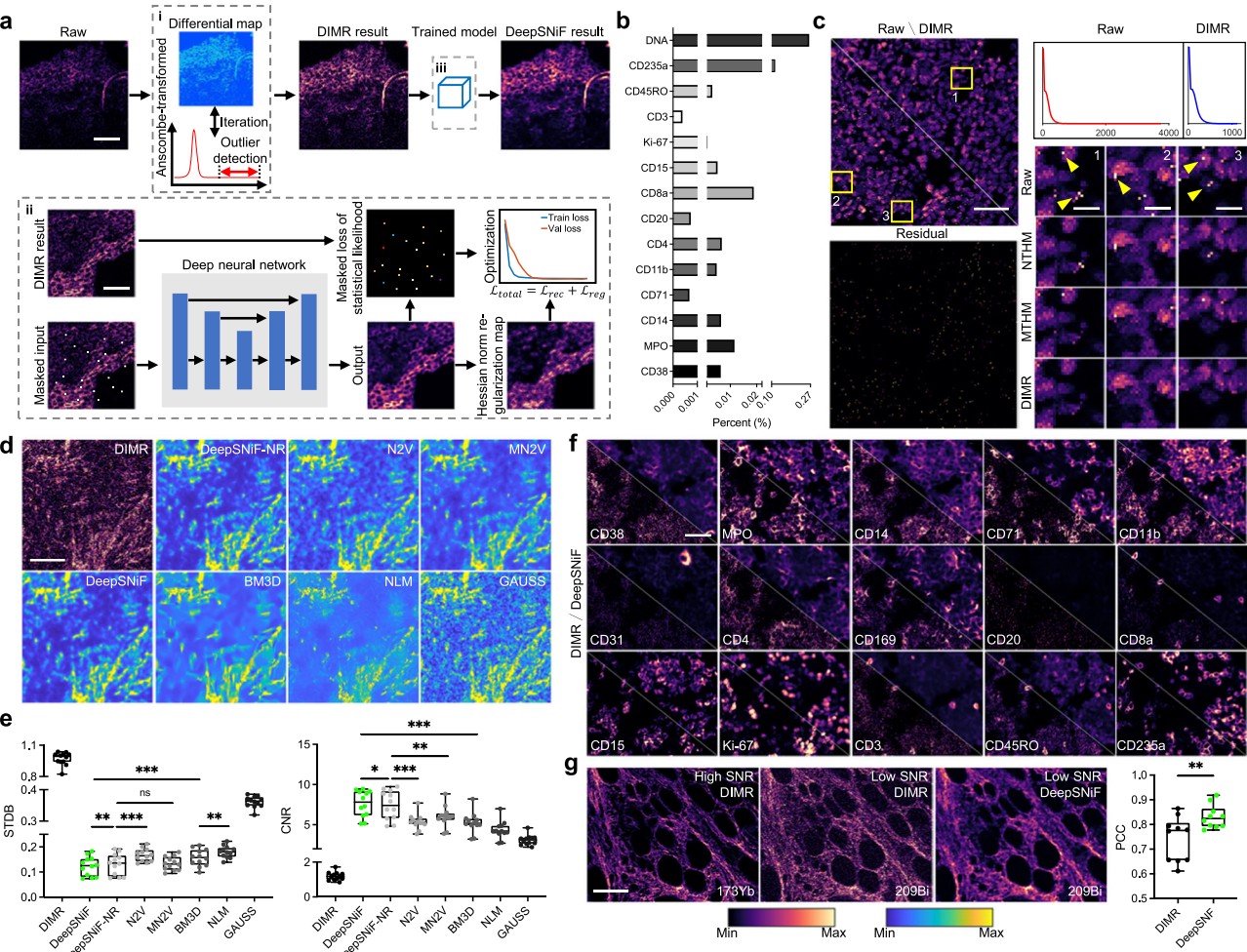

**Fig. 1 | General principle and validation of IMC-Denoise on the human bone marrow IMC dataset. a** Schematic of IMC-Denoise: (i) the DIMR algorithm: after the Anscombe transformation, the difference maps calculated from the raw image are operated to form a histogram. The outliers are detected based on this histogram and removed by a 3 × 3 median filter, iteratively. (ii) The training phase of the self-supervised DeepSNiF algorithm: In the hot pixel corrected images, several pixels are randomly selected and masked. The hot pixel corrected images before and after the masking are set as the outputs and inputs of a deep neural network, respectively. Statistics-derived I-divergence on the masked pixels combined with the Hessian norm regularization on all the pixels is set as the loss function to guarantee the optimal denoising performance. (iii) The prediction phase of the DeepSNiF algorithm: the hot pixel corrected IMC images are fed into the trained network to account for the shot noise. **b** The fractions of detected hot pixels by DIMR in selected channels. **c** DIMR removes hot pixels in DNA intercalator channel effectively. Left: Comparison of the raw and DIMR-processed images; and the

difference between the images, in which Residual corresponds to the detected hot pixels. Upper right: the corresponding histograms of the raw and DIMR-processed images. Lower right: comparisons between the raw, NTHM, MTHM and DIMR processed images. **d** Visual inspection of DeepSNiF and other statistics-based denoising algorithms on a Collagen III-labeled IMC image. **e** DeepSNiF performs significantly better than other algorithms (*n* = 12 independent images) on denoising Collagen III-labeled IMC images in terms of STDB and CNR. **f** Visual inspection of DeepSNiF denoised IMC images labeled with other markers. **g** DeepSNiF improves the Pearson correlations between Collagen III-labeled IMC images with low and high SNR significantly (*n* = 10 independent images). In **e** and **g**, box center indicates median, box edges 25th and 75th percentile, and whiskers minimum and maximum percentile; *P* values were calculated through two-sided Wilcoxon matched-paired test (*P < 0.05, **P < 0.01, ***P < 0.001 and ns: no significance). Scale bar: **a** Upper: 100 μm, lower: 125 μm. **c** Whole region: 75 μm, sub-region 1–3: 8 μm. **d** 50 μm. **f** 45 μm. **g** 100 μm.

adequately remove hot pixels; for a 500 × 500 image, it takes -0.05–0.4 s to run DIMR, depending on the hot pixel densities.

After hot pixel removal, the imaging model is simplified as: $\mathbf{R} = \mathcal{P}[\mathbf{X}]$, for which we have developed DeepSNiF (Fig. 1aii, iii and Supplementary Note 1.4) to account for the ion noise in IMC images. By combining Poisson statistics and detection theory, I-divergence[33] is derived as the loss function to enable the maximized likelihood estimation for the denoising task. Unlike traditional imaging methods for which noise-free training label images can be generated, commonly with long exposures, the image formation process in IMC requires laser ablation. Thus, a tissue can only be imaged once in IMC. Auto-fluorescence artifacts in immunofluorescence (IF) images, and the tedious and potentially interfering processes for consecutive IF and IMC imaging, are further confounds. Therefore, conventional

supervised denoising approaches[34–36] or Noise2Noise[37] are not available here.

We overcome these limitations by applying a self-supervised approach inspired by Noise2Void[24] and Noise2Self[38]. This approach randomly masks several pixels in the DIMR-processed hot pixel-removed images by a stratified sampling strategy. Subsequently, the manipulated images are set as the inputs of the network and the hot pixel removed ones are the outputs. For this construct, the self-supervised training is approximately equivalent to a supervised learning process (Supplementary Note 1.4.1). The network follows U-Net[39] structure with Res-Blocks[40] to enable high quality training and prediction (Methods, Supplementary Fig. 13). Notably, the last activation function of the network is set as softplus (log(1 + exp(x))) to restrict non-negativity for the images. Nevertheless, the denoising

performance is still sub-optimal, due to neglected information of the masked pixels and partially utilized pixels in the self-supervised strategy. To further boost DeepSNiF, the Hessian norm regularization[41–43] is applied in the loss function with the continuity between biological structures a priori (Supplementary Note 1.4.2). Overall, the loss function of DeepSNiF is summarized as Eq. (2).

$$\mathcal{L}(\mathbf{R}, \mathcal{F}_\theta[\,f(\mathbf{R})]) = \sum_p \mathbf{M}_p \cdot \left[ r_p \log \frac{r_p}{\mathcal{F}_\theta[f(\mathbf{R})]_p} - r_p + \mathcal{F}_\theta[\,f(\mathbf{R})]_p \right] \Big/ \sum_p \mathbf{M}_p$$
$$+ \lambda_{\mathrm{Hessian}} \sum_p ||\mathcal{R}_{\mathrm{Hessian}}(\mathcal{F}_\theta[f(\mathbf{R})])||_p \Big/ \sum_p ,$$

(2)

where $\mathcal{F}_\theta$ represents the learnt weights of the network, $f$ demonstrates the random pixel masking approach, $r_p$ the $p$-th pixel of the hot pixel removed training set $\mathbf{R}$, $\mathbf{M}_p$ the pixel mask ($\mathbf{M}_p \in \{0, 1\}$), $\mathcal{R}_{\mathrm{Hessian}}$ the Hessian operator, $\lambda_{Hessian}$ the regularization parameter and $p$ the pixel index. Here, the pixel $p$ is masked only when $\mathbf{M}_p = 1$. Note that the first term works only on the selected masked pixels, while the second regularization term utilizes all of the image information. Prior to training and prediction, the images are normalized between 0 and 1 by a percentile-normalization approach (Supplementary Note 1.4.3). The DeepSNiF algorithm is summarized as Supplementary Algorithm 2. As validated by simulation (Supplementary Note 3.4.2), $\lambda_{\mathrm{Hessian}}$ is empirically set as 3e-6 to balance the trade-off between data fidelity and regularization.

**Validation of IMC image quality improvement**

We initially tested our DIMR algorithm on selected markers of a human bone marrow dataset. Here, inherently high autofluorescence and tissue features (fragile haematopoietic stroma intermixed with dense cortical bone) excluded other spatial biology methods, even after substantial pre-processing. Figure 1b enumerates the proportion of hot pixels detected by DIMR for each marker. We then selected DNA intercalator and CD235a (Fig. 1c and Supplementary Fig. 14) to evaluate DIMR due to their high hot pixel density. By comparing the images and the corresponding histograms, hot pixels are effectively eliminated by DIMR.

We further compared DIMR with two recent hot pixel removal methods, neighbor-based threshold hot pixel removal method (NTHM)[14,15,20] and median-based threshold hot pixel removal method (MTHM)[10] with default parameters, to benchmark its performance (Table 1, Supplementary Note 2.1, Supplementary Algorithms 3 and 4). From the results, all three methods can remove spurious signal, but their performances varied from each other. To quantitatively evaluate these methods, we utilized $t$-CyCIF data[44] to generate simulated IMC images (Supplementary Note 3.1) with a range of noise levels and hot pixel densities. The three methods were then applied on the simulated

datasets, and root mean squared errors between the hot pixel-free and processed images were set as the metric to evaluate the accuracy of hot pixel removal (Supplementary Note 3.2). Note that in simulations, the thresholds of NTHM and MTHM were manually tuned to guarantee their optimal performances, while DIMR was configured automatically.

The simulation results indicate DIMR is the best performer among the three methods (Supplementary Note 3.3.2). In fact, the threshold of NTHM requires contextual adjustment as different tissues and channels may have different scales. Moreover, this method is not efficient at removing consecutive hot pixels. MTHM is not locally adaptive and may overlook hot pixels with similar intensity to that of normal pixels; or erroneously remove normal pixels located at the border between tissues and background. Use of a lower search range or threshold for MTHM may also generate false negatives. In comparison, the outlier detection of DIMR is based on overall image statistics. Therefore, no manual threshold adjustment is required for images with different intensity scales, and a higher detection sensitivity is achieved even for hot pixels with lower intensities. These features along with the simulation data results demonstrate the versatility and accuracy of DIMR. The automated DIMR approach also results in the additional benefit of moderately improved cell segmentation, the result of robust removal of artifacts caused by hot pixels (Supplementary Fig. 15).

With hot pixels removed from image data, we next benchmarked the denoising performance of DeepSNiF along with DIMR and other statistics-based methods including a Gaussian filter with standard deviation of 1 (GAUSS), NLM, BM3D, N2V, modified N2V (MN2V) and DeepSNiF with no regularization (DeepSNiF-NR) (Table 1, Supplementary Notes 2.2 and 2.3) on the simulated dataset (Supplementary Note 3.4). These comparisons were carried out on IMC images labeled with Collagen III, CD31, CD34 and CD3 from the human bone marrow dataset (Fig. 1d and Supplementary Fig. 16). First we visually assessed images with different processing approaches for their overall appearance and in particular for retention of fine cell-level details. We found all the algorithms enhanced the DIMR data even though variant performances were achieved. GAUSS lowers the noise level by sacrificing resolution. NLM is effective at background denoising but does not account adequately for the noise components of signal. BM3D improves NLM further by its cooperative denoising procedure. However, we found it tended to over-smooth foreground and distorted cell shapes. N2V always generates artifacts because of an inappropriate noise model. DeepSNiF-NR performs better than MN2V because the Anscombe transformation in MN2V may generate some bias for extremely low counts; both of which are better than GAUSS, NLM and BM3D. DeepSNiF further enhances these results by mitigating the discontinuities in the DeepSNiF-NR output, and furthermore retains cell morphology features.

We then quantitatively compared the differently processed images across a range of different characterization methods. Assessment of peak SNR (PSNR) and structural similarity (SSIM)[45] (Supplementary

**Table 1 | Reference denoising algorithm summary**

| Acronym | Full name | Algorithm details |
|---|---|---|
| NTHM | Neighbour-based threshold hot pixel removal method[14,15,20] | Supplementary Note 2.1 |
| | | Supplementary Algorithm 3 |
| MTHM | Median-based threshold hot pixel removal method[10] | Supplementary Note 2.1 |
| | | Supplementary Algorithm 4 |
| N2V | Noise2Void[24] | Supplementary Note 2.2.1 |
| MN2V | Modified Noise2Void with Anscombe transformation[27] | Supplementary Note 2.2.2 |
| N2T | Noise2True[34], only simulations used | Supplementary Note 2.2.3 |
| GAUSS | Gaussian filter with kernel size of 5x5 and standard deviation of 0.8 | Supplementary Note 2.3.1 |
| NLM | Non-local means algorithm[22] | Supplementary Note 2.3.2 |
| BM3D | Batch-matching and 3D filtering algorithm[23] with Anscombe transformation | Supplementary Note 2.3.3 |

Note 3.2) were computed from the simulated data, and the standard deviation of background (STDB) and contrast-to-noise ratio (CNR; "Methods" section) were utilized for the IMC images labeled with Collagen III. All results indicated DeepSNiF enables the optimal denoising performance among these algorithms (Fig. 1e and Supplementary Note 3.4). In particular, the noise level (STDB) decreased by 87% and CNR increased by 5.6-fold after DeepSNiF (0.9938 to 0.1254 and 1.1749 to 7.8065, median value).

We further visually inspect the denoising results of IMC-Denoise on multiple datasets including human bone marrow images (Fig. 1f and Supplementary Fig. 17), human breast cancer (Supplementary Fig. 18), human pancreatic cancer (Supplementary Fig. 19) and a MIBI dataset (Supplementary Fig. 20). Image quality improvements that enhance image interpretation are apparent, for both visual inspection and quantitative assessment, in particular for low SNR channels. Two orthogonal staining approaches were pursued in order to provide further validation of these image quality improvements. Firstly, the same antibody was conjugated to two different metals and co-stained on the same tissue for detection in high and low sensitivity channels, without spillover. IMC-Denoise was employed on the low signal channel (209Bi) and was able to restore the image quality to match the high sensitivity channel (173Yb) with the Pearson correlation coefficient (PCC) improved as high as 0.16, as shown in representative images (Supplementary Fig. 21 and Fig. 1g). Similar conclusions can also be drawn from other channels with increased PCC by more than 0.48 and 0.35, respectively (Supplementary Fig. 22). Secondly, tissue sections stained with metal-conjugated antibodies (for CD3, CD4, CD61, and CD169) were probed with a fluorophore-conjugated secondary antibody for IF, individually. We then followed IF imaging by ablative-IMC (Supplementary Fig. 23). The additional handling and washing after IF imaging often leads to extremely low remaining metal isotope signal; however, enhancement in image quality can still be observed to restore the image to correlate to the IF. Specifically, the PCC quantitatively verified the image quality improvement of DeepSNiF (CD3: 0.5557–0.7939, CD4: 0.4975–0.7793, CD61: 0.9096–0.9492, and CD169: 0.4481–0.7726).

## IMC-Denoise enhances IMC background noise removal and downstream analysis

We next evaluated the ability of DeepSNiF in IMC-Denoise to remove background noise of IMC images. Visual inspection (Supplementary Fig. 24) reveal DeepSNiF enhances background noise removal of the examples effectively by a single threshold. To fully evaluate the enhancement by DeepSNiF, we manually annotated 15 images labeled with CD34 and 12 IMC images labeled with Collagen III (Fig. 2a). The single threshold-based method and semi-automated Ilastik-based method[21] were applied on both DIMR and DeepSNiF-processed CD34 and Collagen III images (DIMR_thresh, DeepSNiF_thresh, DIMR_Ilastik and DeepSNiF_Ilastik, respectively), and MAUI was only applied on DIMR images (Methods). The results were compared with the manually annotated ground truths (Fig. 2b), and F1 score was set as the accuracy metric to quantitatively assess the results (Fig. 2c). To guarantee the best performance of threshold-based methods and MAUI, a wide range of parameters were tested (Supplementary Figs. 25–27). Note that in threshold-based methods, optimal thresholds from 1 to 4 were selected for individual DIMR-processed images per marker for fair comparison. Nevertheless, the single threshold 1 was selected for all the images per marker for DeepSNiF-processing, without the need of further tuning.

Overlaid masks and F1 scores for both markers indicated DeepSNiF_Ilastik achieves the highest accuracy while DIMR_thresh is the weakest performer (CD34: 0.9143 to 0.4155, and Collagen III: 0.9434 to 0.5378, median value). Surprisingly, DeepSNiF_thresh is a better method for background noise removal than the semi-automated DIMR_Ilastik (CD34: 0.9040 to 0.8716, and Collagen III: 0.9345 to 0.9108, median value), and its F1 score was improved by approximately

twofold compared to DIMR_thresh. We infer that DeepSNiF is capable of unmixing the signal and background, while the shot noise in DIMR images hinders the performances of the Ilastik-based method. MAUI was able to account for the background noise at the cost of false negative generation (CD34: 0.7824 and Collagen III: 0.7305, median value). Furthermore, we have also visually inspected and manually annotated marker images from Supplementary Figs. 16–18. All the results indicate DeepSNiF achieves good background removal performance (Supplementary Figs. 28–30). Indeed, the signal has been unmixed from background through DeepSNiF because we have proved a simple thresholding can remove background accurately (Fig. 2b, c, Supplementary Figs. 28b, 29b, and 30b). These findings support replacement of tedious semi-automated approaches by automated DeepSNiF.

Next, we were curious to evaluate the impact of IMC-Denoise on single-cell profiles. Using segmented cell masks, we extracted the cell intensities of CD38, MPO, CD14, CD71, CD11b, CD4, CD169, CD20, CD8a, CD15, CD3, and CD235a markers for 96232 cells in total (Methods). Please note that segmentation masks were identical for each comparison, using masks generated from DeepSNiF, so that the impact of variability in segmentation algorithms can be neglected. In Supplementary Figs. 31 and 32a, the comparison of the single-cell profiles of raw, DIMR and DeepSNiF data show that DIMR has the potential to correct false positive data, and DeepSNiF corrects all cell profiles. We have also conducted line fitting for the DIMR and DeepSNiF-processed single-cell data and calculated their PCC (Supplementary Fig. 32a). The results indicate DeepSNiF has not changed the single-cell intensity scale nor biased the overall linearity of the data. Furthermore, larger mean positive marker expressions lead to lighter corrections by DeepSNiF (Supplementary Fig. 30b). This follows from the logic that larger ion counts have lower shot noise levels.

Subsequently, we benchmarked the single-cell data from the raw, DIMR, DIMR_Ilastik and DeepSNiF-processed images (Fig. 2d). To achieve this, manual gating approaches with prior knowledge of cell markers were applied to DIMR, DIMR_Ilastik and DeepSNiF on IMC data (Fig. 2e–h). For example, among T cells (CD3-positive, CD14-negative), myeloid (CD11b, CD15) and erythroid (CD71, CD235a) markers should be absent. However, this condition may not hold because: segmentation and staining artifacts are unavoidable, and because hot pixels are present in the raw data. With the presence of shot noise, the single-cell data could be further biased (Supplementary Note 1.2.2). In Fig. 2e, f, the false positive myeloid and erythroid markers decrease slightly after DIMR correction (0.8% and 0.4%). DIMR_Ilastik and DeepSNiF further removed false positive myeloid (3.86% and 6.31%) and erythroid markers (3.52% and 5.83%) after the slight improvement of DIMR. Similarly, among B cells (CD20-positive), myeloid and erythroid markers (CD11b, CD15, MPO and CD235a) should be absent as well. In Fig. 2g and h, the false positive markers decrease slightly after DIMR correction (0.06% and 0.74%). Compared to DIMR, DIMR_Ilastik and DeepSNiF removed more false positive markers (DIMR_Ilastik: 6.41% and 7.86%, DeepSNiF: 5.22% and 8.44%). Overall, as expected DIMR could enhance the single-cell analysis to a limited extent. DeepSNiF and DIMR_Ilastik enable further enhancement, and overall the former achieves better performance than the latter on this task.

To test whether DeepSNiF-based segmented cell masks potentially favor DeepSNiF data, we have extracted the raw, DIMR, DIMR_Ilastik and DeepSNiF-processed single-cell data from DIMR-based cell masks. Both the single-cell data comparisons (Supplementary Fig. 33) and manual gating results (Supplementary Fig. 34) are similar to that from DeepSNiF-based cell masks. Therefore, we infer that our IMC-Denoise pipeline is also robust across different cell segmentation masks.

## DeepSNiF in IMC-Denoise enhances automated cell phenotyping

Cell phenotype annotation plays a key role in tissue microenvironment analysis. Indeed, false annotation of cell phenotypes has the potential

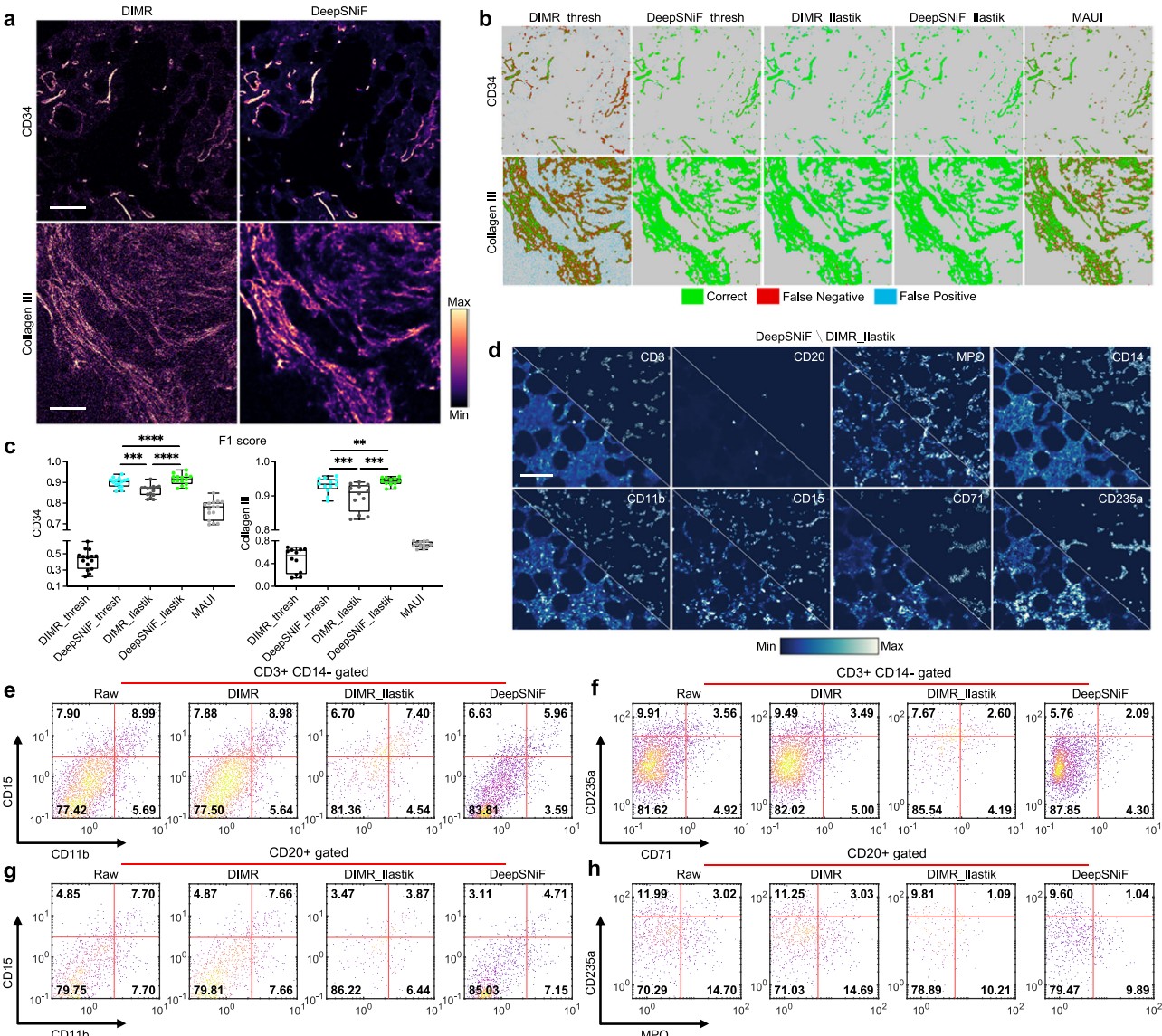

**Fig. 2 | IMC-Denoise enables background noise removal and enhances downstream analysis of the human bone marrow IMC dataset. a** Examples of DIMR and DeepSNiF-processed IMC images labeled with CD34 and Collagen III. **b** Visual inspection of background removal results of DIMR and DeepSNiF-processed images, in which DIMR_thresh and DeepSNiF_thresh are binarized with the optimal thresholds (Supplementary Figs. 25 and 26), DIMR_Ilastik and DeepSNiF_Ilastik are segmented by the Ilastik software package, and MAUI results are the DIMR images processed by the MAUI software package (Supplementary Fig. 27), respectively. Manual annotated images are served as ground truths. **c** After DeepSNiF denoising, the background removal accuracy improves significantly in terms of F1 score, for both CD34 and Collagen III-labeled images ($n = 15$ independent images for CD34 and $n = 12$ independent images for Collagen III). Notably, DeepSNiF_Ilastik achieves

the highest accuracy, while DeepSNiF_thresh performs better than all the background removal results from DIMR images. Box center indicates median, box edges 25th and 75th percentile, and whiskers minimum and maximum percentile. $P$ values were calculated through two-sided Wilcoxon matched-paired test (**$P < 0.01$, ***$P < 0.001$, and ****$P < 0.0001$). **d** Visual inspection of DeepSNiF and DIMR_Ilastik-based denoising results on different markers-labeled IMC images. **e–h** Evaluations of denoising algorithms with manual gating strategies on single-cell data. The numbers in these panels are the cell percentages of the corresponding ranges. DIMR slightly enhances the single-cell analysis over raw data, while DeepSNiF further enhances the DIMR results and overall performs better than semi-automated DIMR_Ilastik-processing. Scale bar: **a** Top: 50 μm, bottom: 35 μm. **d** 107 μm.

to lead to false biological or clinical conclusions. Hot pixel removal is normally conducted before automated cell phenotyping[14,15,17]. Therefore, we focused on whether DeepSNiF in IMC-Denoise could impact phenotypic annotation of cell types. Here, the extracted single-cell data with DeepSNiF-based segmentation masks from the human bone marrow dataset were used for phenotypic annotation, including CD38, MPO, CD14, CD71, CD11b, CD4, CD169, CD20, CD8a, CD15, CD3 and CD235a channels. We clustered the DIMR dataset by the Phenograph algorithm[46] with the Leiden community detection algorithm[47] (Methods). The generated clusters were then annotated as immune cell subsets (B cell, CD4+ T cell, CD8+ T cell and plasma cell, monocyte/

macrophages), erythroid, myeloid, and other CD4+ cells and others. To better demonstrate the modifications of DeepSNiF denoising, we then utilized a weighted KNN approach to map the DeepSNiF data into the DIMR-based clusters (Methods). The weights were acquired by calculating the Jaccard index between each DeepSNiF-processed cell profile with all of the DIMR cells (which is identical to the Jaccard graph construction of Phenograph). For visualization, the cell markers of DIMR and DeepSNiF were also compressed into two dimensions by the fast interpolation-based $t$-SNE algorithm[48] as Supplementary Fig. 35 (Methods). The assigned phenotypes of DIMR and DeepSNiF datasets are demonstrated in Fig. 3a and the relative changes of each cell sub-

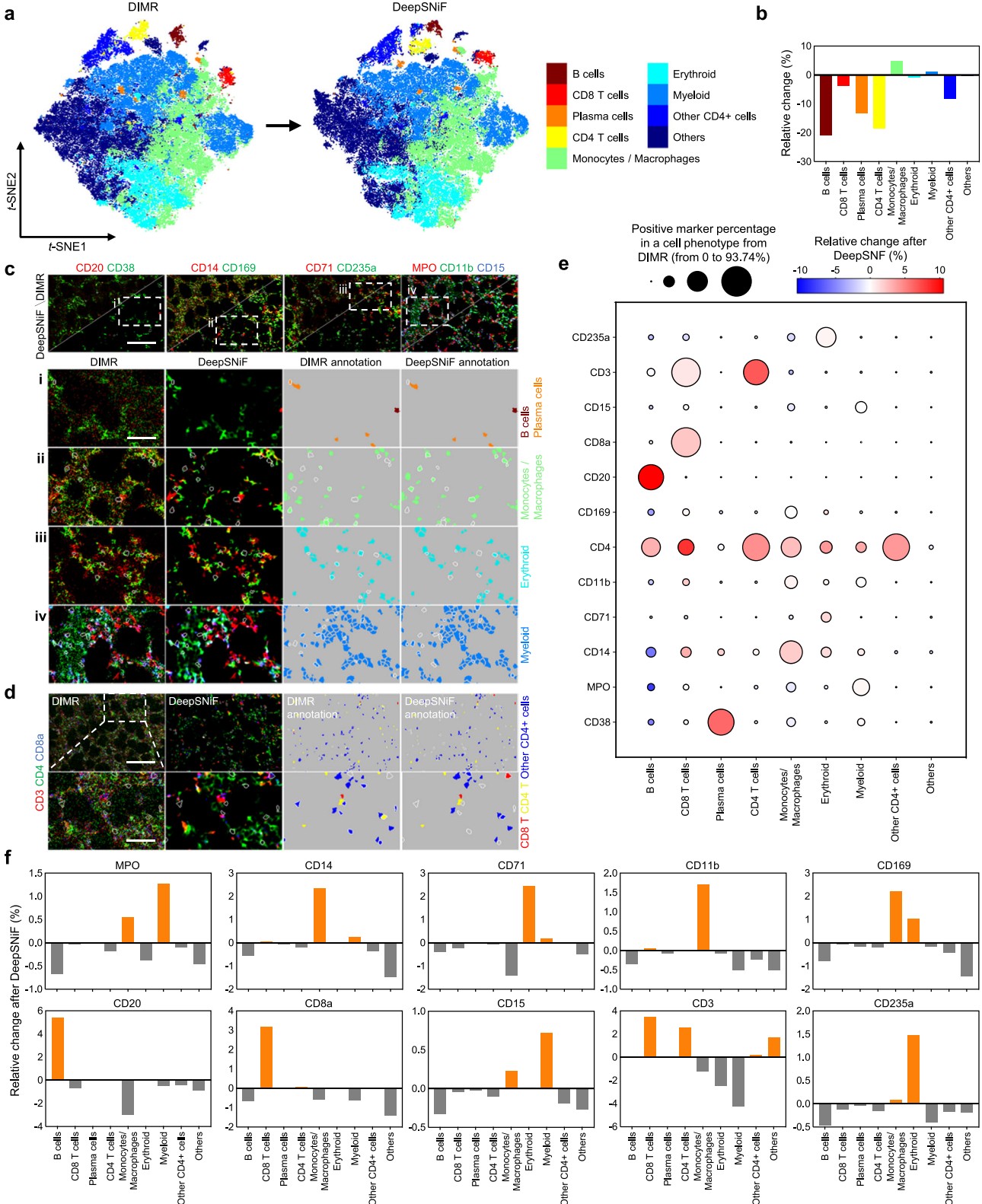

**Fig. 3 | DeepSNiF enhances automated cell phenotyping on human bone marrow IMC data. a** *t*-SNE plots of DIMR and DeepSNiF with cell phenotyping results. **b** The relative change in cell phenotypes before and after DeepSNiF. **c**, **d** Comparisons of DIMR and DeepSNiF-processed IMC images labeled with different cell markers, and the corresponding cell annotation results. The sub-panels (i)–(iv) in **c** and the bottom row in **d** correspond to the white dashed box region selection in their first panels, respectively. The white contours represent the differential phenotyping results between DIMR and DeepSNiF. **e** DeepSNiF enhances the sensitivity of cell phenotyping. After DeepSNiF processing, the non-specific

marker signals reduce while the specific ones enrich in the cell types, respectively. The circle size indicates the positive marker percentage in a particular phenotype of DIMR, and the circle color indicates the relative changes of the positive rate for the particular markers after DeepSNiF enhancement. **f** DeepSNiF enhances the specificity of cell phenotyping. With DeepSNiF denoising, the ratios of specific phenotypes increase while those of non-specific phenotypes decrease in the positive markers. The relative change is the difference in percentage composition of each cell type before and after DeepSNiF enhancement. Scale bar: **c** 110 μm. **d** Top: 145 μm, bottom: 50 μm.

population after DeepSNiF processing is shown in Fig. 3b. After DeepSNiF processing, B cells, CD8 T cells, plasma cells, CD4 T cells and other CD4+ cells decrease (20.86%, 3.70%, 13.44%, 18.49%, and 8.23%, respectively), the monocytes/macrophages increase (4.82%), while erythroid, myeloid and other cells remain largely unchanged.

The phenotyping results of DIMR and DeepSNiF were mapped back into their segmentation masks and images (Fig. 3c, d and Supplementary Fig. 36). To highlight cells where DeepSNiF changes the cell phenotyping results, conflicting annotations between DIMR and DeepSNiF were labeled with white contours, and the changes were quantified for cell phenotype and marker enrichments (Methods). After DeepSNiF denoising, non-specific markers are reduced, while specific markers are enriched within the cell phenotypes (Fig. 3e). For example, we observed the positive rate increased for CD20 in B cells (10.53%), CD8a in CD8 T cells (2.32%), CD3 and CD4 in CD4 T cells (6.84% and 4.64%), CD38 in plasma cells (6.21%) and CD4 in other CD4+ cells (4.26%). Conversely, we observed a decrease of non-specific markers, such as CD38, MPO and CD14 in B cells (5.24%, 8.11%, and 5.64%), CD3 in erythroid (1.79%) and myeloid (1.62%) cells, and all marker signals in "other" cells. Furthermore, the identified cell types were enriched in a marker-specific manner after DeepSNiF (Fig. 3f). For instance, we observed a post-DeepSNiF enrichment of monocytes/macrophages in CD14+ cells (2.36%), CD11b+ cells (1.70%) and CD169+ cells (2.21%), and enrichment of B cells in CD20+ cells (5.42%) and CD8 T cells in CD8a+ cells (3.15%). Similarly, myeloid cells were enriched in MPO+ (1.26%), and erythroid cells in CD71+, CD235a+ cells (2.45% and 1.48%). DeepSNiF also yielded an enriched composition of CD8 and CD4 T cells (3.50% and 2.54%), and reduced composition of myeloid and erythroid cells (2.47% and 4.27%) in CD3+ cells. However, we noticed the enrichment of erythroid cells in CD169+ cells (1.03%), which may result from an artifact of the current segmentation approach due to the close relationship and irregular morphology at the boundaries between erythroids and macrophages within the bone marrow[49].

Cell phenotyping by immunostaining of FFPE tissues is also inherently limited by antibody specificity and antigen retrieval protocols. In this tissue, CD38+ and CD14+ antibody staining is not strictly restricted to single lineages, and these markers can be aberrantly expressed in myeloid neoplasms included in this data set (Supplementary Fig. 37). On manual inspection, DeepSNiF improves the ability to identify co-localization of cell surface markers (Fig. 3c, d and Supplementary Fig. 36). Overall, DeepSNiF enhances the sensitivity and specificity of cell phenotyping. We have conducted similar analysis for the single-cell data from the DIMR-based segmentation masks as well (Supplementary Fig. 38). Likewise, the results are also similar to those from the DeepSNiF-based segmentation masks, which demonstrates the robustness of IMC-Denoise.

We observed that the enhancements in cell phenotyping and marker enrichments in Fig. 3e, f are related to the noise level of the IMC images. Specifically, DeepSNiF has the highest impact on CD20 and CD3 related phenotypes, improvement for CD15, MPO and CD235a related phenotypes is limited, with moderate changes for other cell classes. These findings agree with Supplementary Fig. 32b, where we plot the STD of the normalized positive marker differences between DIMR and DeepSNiF against intensity. To investigate the influence of DeepSNiF on phenotyping results more deeply, we applied a leave-one-out DeepSNiF strategy for CD20, CD3, CD71, CD235a, and MPO ("Methods" section and Supplementary Figs. 39–43). Briefly, this involved processing one marker for hot pixel removal with DIMR, e.g. CD20, and all the other markers by both DIMR and DeepSNiF. Then the same weighted KNN approach was applied on these leave-one-out DeepSNiF datasets. Similar to the conclusions from Fig. 3e, f and Supplementary Fig. 32b, CD20 and CD3 denoised by DeepSNiF improve cell phenotyping because of the high noise level of the corresponding IMC images. DeepSNiF has moderate impact on CD71 due to better IMC image quality than those

of CD20 and CD3, and has minor impact on MPO and CD235a because of their good SNRs.

## DeepSNiF in IMC-Denoise enhances lymphocyte analysis

Cell-cell interactions of immune cells within the tumor microenvironment is of broad interest for many clinical pathology specimens. In myeloid malignancies, immune infiltrates are most commonly assessed by flow cytometry and are an active area of interest in therapeutic clinical trials[50]. However, in situ spatial context of cell-cell interaction mediated immune responses cannot be directly measured through this approach. We quantified the enhancement of lymphocyte spatial analysis for B cells, CD8+ T cells and CD4+ T cells by DeepSNiF, and compared these to a manually curated set of image annotations based on DeepSNiF-based cell masks (Fig. 4a). CD3, CD4, and CD20-stained images are more easily contaminated by shot noise than others (Supplementary Fig. 32b). Therefore, this approach can further validate the shot noise accounting ability of DeepSNiF as well. The phenotyping accuracy of DIMR and DeepSNiF as evaluated by the Jaccard score and F1 score indicate a significant improvement by DeepSNiF denoising (Fig. 4b); and DeepSNiF denoised data closely recapitulates gold-standard but laborious manual annotation. Specifically, the overall Jaccard scores improve from 0.6785, 0.8229, and 0.6781 to 0.9201, 0.8922, and 0.8860 for B cells, CD8+ T cells and CD4+ T cells, respectively. Similarly, the F1 scores improve from 0.8085, 0.9029, and 0.8082 to 0.9584, 0.9430, and 0.9396 for these cell types, respectively. We have also compared the annotation results on the DIMR-based cell masks (Supplementary Fig. 44). While there are some variations due to the differences from segmentation masks, this comparison demonstrates accuracy improvements as well.

Subsequently, the tissues were classified as normal morphology (Normal), myelodysplastic syndromes (MDS) and acute myeloid leukemia (AML). The improvements in visual quality afforded by DeepSNiF denoising facilitated manual review of lymphocyte staining patterns for annotation annotation of lymphocyte subpopulations (Supplementary Fig. 45). B and T cell populations are scattered throughout the bone marrow cellularity in normal and malignant specimens, and lymphoid aggregates are occasionally present (Fig. 4c). To characterize the density and distance relationships between lymphocyte subpopulations, samples were analyzed in cohorts of extent of malignant blast involvement, after exclusion of the lymphoid aggregate outlier. Nearest neighbor distances between B cells, CD4 T and B cells, and CD4 and CD8 T cells were calculated for different disease tissues (Fig. 4d). Overall, the distributions from DeepSNiF are more concordant with annotated data. By contrast, those from DIMR are biased, with significant differences to the annotations due to cell misclassifications.

Automated DeepSNiF denoising reveals that as disease develops, the B cell distances become larger ($P < 0.01$); The distances between CD4 T and B cells in normal and MDS tissues are greater than those of AML ($P < 0.0001$); And CD4 and CD8 T cells in normal tissues trend towards longer distances than those in MDS ($P = 0.0916$). Interestingly, the overall distances between CD4 T and B cells in MDS tissues are greater than those of Normal samples ($P < 0.01$). These findings hold for DeepSNiF denoised data in distances between B cells from MDS to AML ($P < 0.05$), and from Normal to MDS samples for CD4 and CD8 T cells ($P = 0.0929$) and CD4 T to B cells ($P < 0.05$). However in non-DeepSNiF denoised DIMR data, the trends between B cells has been violated from MDS to AML by DIMR data ($P = 0.4923$), as well as those from Normal to MDS between CD4 and CD8 T cells ($P = 0.4762$), and CD4 T to B cells ($P = 0.6685$). From this point, DeepSNiF is able to correct the distorted cell spatial distributions from less accurate annotations caused by noise, which may further enhance downstream cell-specific spatial analyses.

We also calculated the cell densities per tissue of these lymphocytes (Fig. 4e). The DeepSNiF results are closer to those annotated data

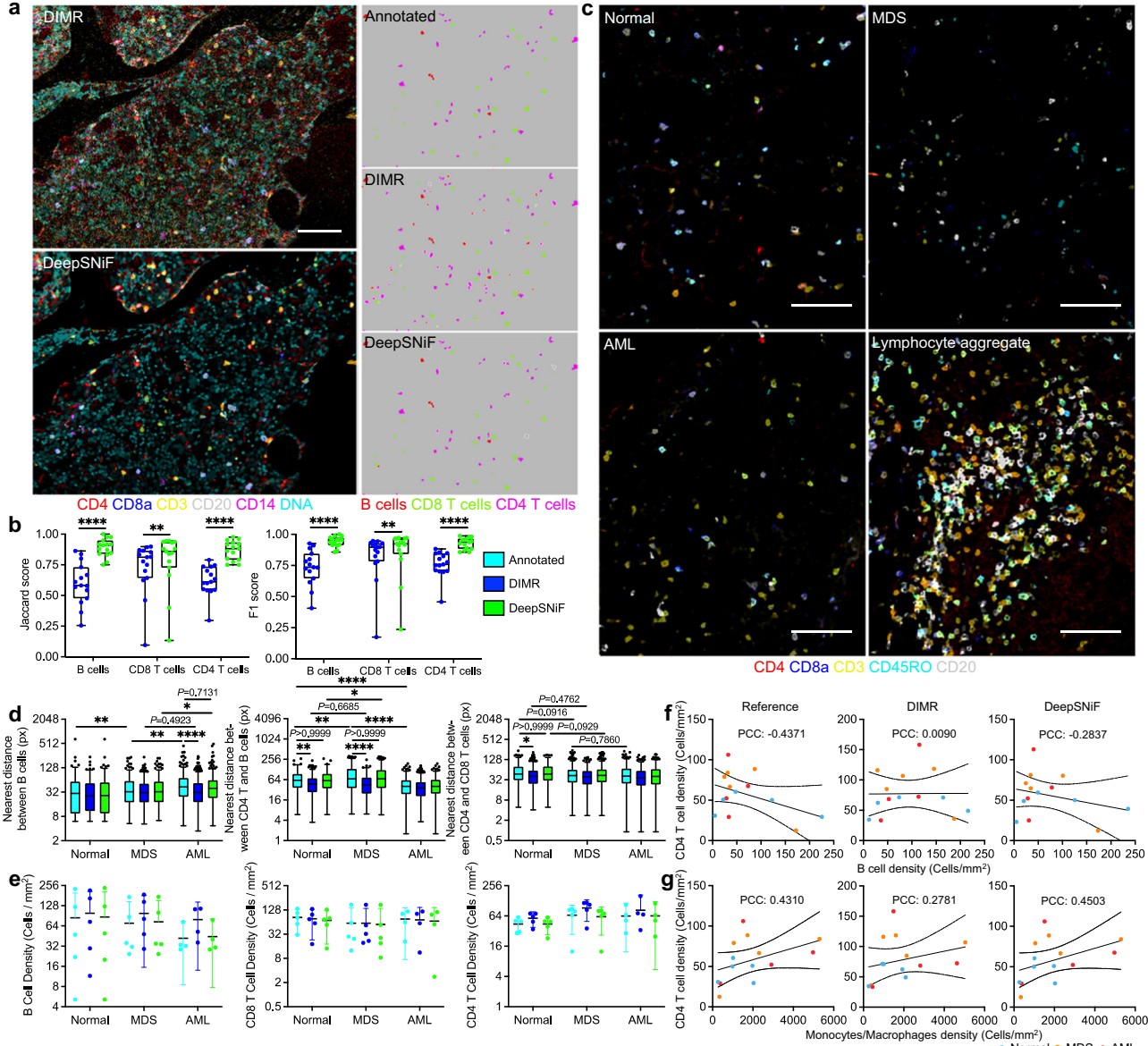

**Fig. 4 | DeepSNiF enhances lymphocyte analysis. a** Manual annotations for lymphocytes and comparisons with DIMR and DeepSNiF phenotyping results with DeepSNiF-based cell masks. The white contours represent the differential phenotyping results between the annotated and DIMR/DeepSNiF results. **b** Annotation evaluations of DIMR and DeepSNiF by both Jaccard and F1 scores across all the tissues ($n = 15$ biologically independent samples). Box center indicates median, box edges 25th and 75th percentile, and whiskers minimum and maximum percentile. $P$ values were calculated through two-sided Wilcoxon matched-paired test (**$P < 0.01$ and ****$P < 0.0001$). **c** Representative images of lymphocyte markers after DeepSNiF denoising from specimens of normal (upper left), myelodysplastic syndromes (MDS, upper right), acute myeloid leukemia (AML, lower left) and AML with lymphoid aggregate (lower right) tissue samples. **d** Nearest distance comparisons between different cell types of normal ($n = 5$ biologically independent samples), MDS ($n = 5$ biologically independent) samples and AML ($n = 4$ biologically independent samples) tissues from manual, DIMR and DeepSNiF phentyping results. Tukey box center indicates median, box edges 25th and 75th percentile, and whiskers the highest and lowest values that are not outliers. Outliers (single points) are defined as values that are more than 1.5 times the interquartile range). **e** Cell densities comparisons of normal ($n = 5$ biologically independent samples), MDS ($n = 5$ biologically independent samples) and AML ($n = 4$ biologically independent samples) tissues from manual, DIMR and DeepSNiF phentyping results. The center bars define the mean, and the error bars are 95% confidence interval. $P$ values were calculated through two-sided Kolmogorov-Smirnov test (*$P < 0.05$, **$P < 0.01$, and ****$P < 0.0001$). **f, g** Correlation analysis between CD4 T cell and B cell, monocyte/macrophage densities per tissue from manual, DIMR and DeepSNiF phentyping results. The data from the reference group in **f** comes from annotated data; while that from the reference in **g** comes from annotated (CD4 T cells) and DeepSNiF (monocytes/macrophages) results, separately. The solid lines represent the fitting results with the data while the dashed lines represent 95% confidence interval. Scale bar: **a** 85 μm; **c** 112 μm.

for B cell and CD4 T cell. By contrast, the CD8 T cell densities from DIMR, DeepSNiF and annotated data are close to each other. Additionally, we observe higher B cell and CD4 T cell densities in Normal tissues than others, and higher B cell density of MDS than that of AML. No obvious developing trend for CD8 T cells are observed as the disease status changes. Furthermore, we have analyzed the correlations between the densities from different cell types (Fig. 4f, g). Note that in

the reference groups, the B and CD4 T cell densities are generated from the annotated data, while the monocyte/macrophage density comes from DeepSNiF data. This is because the relative change of the monocytes/macrophages by DeepSNiF is smaller compared to those of B and CD4 T cells (Fig. 3b) and because DeepSNiF achieves higher accuracy than DIMR for the cell phenotyping (Fig. 3e, f). From the reference group in Fig. 4f, the densities of CD4 T and B cells are

negatively correlated with each other (PCC: -0.4371). Nevertheless, DIMR result indicates no correlation between the cell densities (PCC: 0.0090), which demonstrates false annotations hinder true relational definition between different cell types. Again, the negatively correlated relationship can be uncovered using automated DeepSNiF (PCC: −0.2837). Likewise, the DIMR data fails to detect the correlation between the densities of CD4 T cell and monocyte/macrophages (PCC: 0.2781; Fig. 4g). Corrected by DeepSNiF, the measured correlation (PCC: 0.4503) approximates the reference finding (PCC: 0.4310).

## Discussion

With the rise of novel multiplexed technologies for the characterization of cellular context in health and disease, IMC has emerged as a valuable tool to investigate immunophenotypes while preserving spatial information[8–16]. Differing from traditional multiplexed imaging approaches based upon fluorescence microscopy, IMC allows for simultaneous acquisition of more than 40 cell-specific markers with greatly suppressed channel crosstalk, and avoids tissue and marker degradation in multi-round staining protocols. Furthermore, it eliminates autofluorescence and background signal issues that are inherent in fluorescent microscopy. The high-dimensional datasets then enable complex microenvironment analysis. However, IMC suffers from unique hot pixel and shot noise features. Analyzing raw IMC data without further restoration may lead to distortions, even errors, in downstream analysis. Contemporary denoising strategies[10,14,15,19,21] are usually not adaptive or effective for these particular noise conditions. For example, the parameters of some methods must be tuned manually, which is not suitable for large datasets and may cause subjective, batch, and channel-specific errors.

In this work, we propose IMC-Denoise to account for the specific technical noise present in IMC images. In this pipeline, the DIMR algorithm is first applied to adaptively remove hot pixels. It does not use a user-defined intensity threshold or range to define hot pixels, eliminating the impact of the density and intensity variations of hot pixels in different datasets or markers. Instead, it builds a histogram from the differential maps of raw images followed by an iterative outlier detection algorithm. In comparison with other methods, DIMR achieves more robust hot pixel detection capability and normal pixel preserving performance. After hot pixel removal, the DeepSNiF algorithm is proposed to restore image quality. I-divergence is derived as the optimal loss function for this denoising task. Due to the absence of noise-free IMC images and incompatibility with repeated scanning to generate training labels[8], we applied a masking strategy with stratified sampling from Noise2Void[24]. This enabled self-supervised training for this denoising task, in which multiple pixels are randomly masked and replaced by its adjacent pixels. With the continuity of antibody signals in IMC, Hessian norm regularization[41–43] is added in the loss function to boost the denoising performance. In DeepSNiF, we train a single network for a single marker, which reduces the memory allocated for training. Nevertheless, we note that DeepSNiF also works on multi-marker training (Supplementary Fig. 46). In another aspect, this demonstrates that DeepSNiF works on the markers stained for morphologically heterogeneous markers, since the variant features have been learned in the training process. In addition, monocytes/macrophages are morphologically heterogeneous so that the successful denoising of CD14/CD169 (Fig. 1f) validates the adaptability of DeepSNiF as well. In fact, the networks are able to learn all the features existing in the training images but not focus on any specific structures. As a result, markers with interstitial staining patterns (e.g. vessels, fibrosis, reticular cytoplasmic projections) can be well restored (CD31, CD34 and Collagen III in Figs. 1 and 2 and Supplementary Figs. 19 and 20). However, small areas of staining at the size of a sub-cellular synapse (e.g. 1–2 μm diameter) will not be successfully distinguished by IMC due to its relatively low resolution of 1 μm. Therefore, the network cannot learn the features of such small structures. The trained

network can be employed to other datasets which share similar features (Supplementary Fig. 22).

To determine the applicability of our approach, reference denoising algorithms were utilized to rigorously evaluate IMC-Denoise on both simulated data and multiple pathological patient datasets. Compared to other methodologies, both DIMR and DeepSNiF achieve the best denoising performance, qualitatively and quantitatively. Orthogonal approaches that have not been previously tested in evaluation of IMC restoration are also used to verify the image quality improvement by IMC-Denoise. This pipeline can be further extended by existing analytical processing pipelines including Mesmer/DeepCell and ark-analysis[25] or MCMicro[26]. If warranted, one may[18] address spillover issues after hot pixel removal and shot noise filtering, as indicated in Eq. (1). A related modality, MIBI[6,7], shares several image formation and noise features with IMC, and the denoising pipeline deployed here may also enhance MIBI datasets.

IMC-Denoise is effective at removing background noise and enhancing downstream analysis of IMC data with limited, subjective, user-input. Multiple datasets processed by DIMR and DeepSNiF were compared with state-of-the-art IMC background removal methods, including single threshold binarization, semi-automated Ilastik-based[21], and MAUI[19], using the F1 score as the accuracy metric to evaluate the results. The qualitative and quantitative results indicate DeepSNiF can affect significant background noise removal, and is superior to tedious semi-automated approaches. In particular, DeepSNiF is capable of unmixing specific IMC staining signal from background noise. This means that even the thesholding approach for background removal is not essential after DeepSNiF denoising.

Conventional workflows typically use manual gating strategies combined with prior cell marker knowledge to identify and compare cell types in pathological samples. We used real world data and these methods to evaluate the IMC denoising algorithm for single-cell analyses, and compared to DIMR, DIMR_Ilastik, and DeepSNiF. Automated IMC-Denoise performs equally or superior to the semi-manual Ilastik-based method in downstream single cell analysis, and DeepSNiF notably enhances cell clustering and annotation. Quantitative evaluations of cell phenotyping results indicate the improvement of sensitivity and specificity by DeepSNiF denoising. Further validations with DIMR-based cell masks demonstrate the robustness of IMC-Denoise to variant cell segmentation results. For lymphocyte annotation, Jaccard and F1 scores demonstrate that DeepSNiF performs significantly better than DIMR on phenotyping of B, CD8-positive T and CD4-positive T cells. Further, spatial distribution and cell density correlation analysis indicate less accurate annotations by the data denoised solely by DIMR, leading to biased conclusions. With the data denoised by DeepSNiF, such distortions can be corrected and more accurate downstream analysis is achieved.

As noted, DeepSNiF enhances all the markers and their downstream analysis. However, the marker channels with high noise levels benefit to a larger degree. In theory, there is no maximum noise level present for denoising algorithms. Even under some extremely noisy conditions (CD20 and CD3 in Fig. 1f, markers in Supplementary Figs. 18 and 20), DeepSNiF improves the image quality. Nevertheless, lower SNR in raw images means lower specific information content and thus the quality of the restored images are lower (Supplementary Figs. 6–11). Because of the signal-dependent characteristics of shot noise, the noise components of high SNR channels contribute less to overall image quality, and thus have lower impact on downstream analysis. Empirically, we find that denoising by DeepSNiF can be omitted when the mean expressions of positive markers are larger than 7 (MPO, CD15, and CD235a), however denoising all marker channels improves performance and is not computationally intensive.

Limitations of IMC-Denoise include the inability to remove large hot pixel clusters, as DIMR cannot discriminate these larger areas of outliers from signal (Supplementary Fig. 47). Further the self-

supervised DeepSNiF algorithm cannot reach the accuracy of supervised denoising methods due to unavailability of ground truths (Supplementary Figs. 8 and 9). Nevertheless, DIMR can remove single hot pixels and small hot clusters of several consecutive pixels, and DeepSNiF performs better than other unsupervised and self-supervised denoising methods on IMC datasets. To conclude, we have developed the content aware IMC-Denoise for improved IMC image quality and quantitative accuracy. Predicated on a novel combination of differential map-based and self-supervised CNN-based algorithms, this open source pipeline removes hot pixels and effectively suppresses shot noise in multiplexed IMC data. Multiple image and cell-based analyses from different IMC datasets verified the enhancements brought by this approach, with the ability to resolve significant cellular phenotypic and spatial information approximating manual annotation. We have provided tutorials to help users implement IMC-Denoise (Supplementary Note 4 and https://github.com/PENGLU-WashU/IMC_Denoise). We expect IMC-Denoise to become a widely used pipeline in IMC analysis due to its adaptability, effectiveness and flexibility.

## Methods

### Human bone marrow dataset

Sections were cut in 4-6 μm thickness from formalin-fixed paraffin-embedded (FFPE) blocks of ethylenediaminetetraacetic acid (EDTA)-decalicifed bone marrow trephine biopsy specimens. Three patients demonstrated normal morphology, and 4 patients were diagnosed with myelodysplastic syndromes (MDS), with additional later time-points obtained at disease progression including acute myeloid leukemia (AML). Use of specimens for secondary analysis in this study was approved by the Washington University in St. Louis Institutional Review Board (#201912110). Informed consent was waived, per IRB-approved protocol.

### Tissue staining and IMC data acquisition

Descriptions of cell markers and isotope tags are provided in Supplementary Tables 2–5. Staining was performed according to Fluidigm IMC recommendations for FFPE as follows. Briefly, tissue sections were dewaxed in xylene and rehydrated in a graded series of alcohol. Epitope retrieval was conducted in a water bath at 96 °C in Tris-EDTA buffer at pH 9 for 30 minutes, then cooled and washed in metal-free PBS. Blocking with Superblock (ThermoFisher) plus 5% FcX TruBlock (Biolegend) was followed by staining with antibody cocktail prepared in 0.5% BSA and metal-free PBS overnight at 4 °C. Sections were washed in 0.02% TritonX100 followed by metal-free PBS, then nuclear staining was performed using 1:200 or 1:300 dilution of Intercalator-Ir (125 μm, Fluidigm) solution for 30 min, followed by ddH$_2$O for 5 min. Slides were air-dried before IMC measurement.

The abundance of bound antibody was quantified using the Hyperion imaging system (Fluidigm) controlled by CyTOF Software (version 7.0.8493), with UV-laser set at 200 Hz. Count data were then converted to tiff image stacks for further analysis using MCD Viewer (version 1.0.560.6, Fluidigm) or imctools (Bodenmiller lab, https://github.com/BodenmillerGroup/imctools).

### Tissue staining and IF data acquisition

For IF staining, tissue was prepared using the same protocal in IMC staining, then stained overnight at 4 °C with a single metal-conjugated primary antibody (CD3, CD4, CD169, or CD61 in Supplemental Table 5). The single-stained tissue was washed, then stained with secondary antibody (donkey anti-rabbit AF647 or goat anti-mouse AF750, Invitrogen, 2 mg/mL diluted 1:400 in 0.5% BSA in PBS) at room temperature 1 h, followed by additional washing in PBS and DAPI (1 μg/mL) staining. Slides were mounted with SlowFade Glass antifade reagent (ThermoFisher) and # 1 1/2 coverslips. Images were acquired using Leica DMi8 inverted widefield microscope with Lumencor SOLA SE U-nIR light engine, DAPI/FITC/TRITC/Cy5/Cy7 filters, DFC9000 GT

sCMOS camera, PL APO 20x/0.80 objective and LAS X software (version 3.7.3.23245). After image acquisition, coverslips were removed with gentle agitation in PBS, then Ir-intercalator staining, washing and drying performed as above for subsequent Hyperion data acquisition.

### Human pancreatic, breast cancer IMC datasets, and MIBI dataset

We applied the human pancreatic[10], breast cancer[12] IMC datasets, and a MIBI dataset[19] to verify the flexibility of IMC-Denoise. All of these datasets are publicly available. The links are provided in the corresponding papers. In breast cancer dataset, CD3, CD20, CD45, CD68, c-Myc, EGFR, EpCAM, Ki-67, Rabbit IgG H L, Slug, Twist, and vWF were selected; in pancreatic cancer dataset, CD3, CD4, CD8, CD11b, CD14, CD31, CD44, CD45, CD45RO, CD56, Foxp3, and pS6 were selected; and in the MIBI dataset, CD3, CD4, CD8, CD11b, CD11c, CD14, CD20, CD31, CD45, CD68, CD206, and HLA-DR were selected. The two IMC datasets were processed by both DIMR and DeepSNiF. The MIBI dataset is only processed by DeepSNiF because no hot pixels are observed, and the hot clusters observed in MIBI images can be removed by the MAUI software package[19]. Details on software implementation can be found in the relevant sections below.

### Neural network implementation

The DeepSNiF neural network follows the U-Net architecture[39] with Res-block modules[40], in which the input and output images share the same size (Supplementary Fig. 13). U-Net architecture is widely used for image deblurring and denoising[24,34]. In general, the network is composed of an encoder and a decoder. Starting with the input, the encoder path gradually condenses the spatial information into high-level feature maps with growing depths; the decoder path reverses this process by recombining the information into feature maps with gradually increased lateral details. The information in adjacent feature maps transfers by convolving with 3 × 3 convolutional filters. The down-sampling and up-sampling are used in encoder and decoder for compressing and reconstructing features, performed here by 2 × 2 max-pooling and 2 × 2 up-sampling operations, respectively. Res-blocks are applied to facilitate efficient training. Each res-block contains a convolution layer, batch normalization and the rectified linear unit (ReLU) nonlinear activation, in which the batch normalization layer aims to speed up training process, ReLU could provide non-linearity in the network. Drop out layers are also added with 0.5 dropout rate after the central two res-blocks to mitigate overfitting. The skip connections link low-level features and high-level features by concatenating their feature maps. We use the softplus function ($\log(1 + \exp(x))$) as the activation function of the final layer and Eq. (2) as the loss function so that the output of the network is guaranteed to be non-negative.

The hot pixel-removed images are split into multiple 64 × 64 patches. Then, the patches are rotated by 90°, 180°, and 270°, and flipped as a data augmentation approach. In IMC images, foreground objects of interest might be distributed sparsely. In this case, the model might overfit the background areas and fail to learn the structure of foreground objects if the entire image is used indiscriminately for training. Therefore, patches from the background regions are excluded from training. In IMC images, pixels with intensity value 0 are considered as background. Afterwards, we define the background pixel ratio $r$ as ratio of the number of background pixels and that of total pixels in the patch. Patches are considered as the background regions if $r \le \rho$, where $\rho$ is the threshold and set from 0.2 to 0.99 for different channels and datasets. We applied a smaller $\rho$ for the datasets less sparse images and vice versa. For good generalization ability of the network, we recommend at least 5000 patches for training. Before training, all the generated patches were percentile normalized (99.9–99.999, Supplementary Note 1.3.3). The percentile of 99.9 was applied for those training sets with extremely bright markers and larger percentile with relatively homogeneous intensity distributions. To

balance the training efficiency and accuracy, 0.2% pixels of each patch are masked and replaced by their neighbors using a stratified sampling strategy[24]. Finally, 85% of the patches are set as training set and the rest as validation set.

All models were trained using Keras[51] (version 2.3.1) on a single NVIDIA Quadro RTX 6000 GPU with 24 GB of VRAM. Adam optimizer[52] was applied as the optimization algorithm with a initial learning rate of 0.001 for 200 epochs and batch size of 128. Learning rate is multiplied by 0.6 if validation loss does not improve for 20 epoches. The training details for all the datasets are summarized as Supplementary Tables 6–11. Note that the training datasets for N2V, MN2V, and DeepSNiF-NR are the same as those for DeepSNiF, and the training time of N2V, MN2V, and DeepSNiF-NR is approximately equal to that of DeepSNiF.

### Neural network inference details
Given a trained denoising model, we denoise full-size IMC images to avoid edge stitching effects. In order to achieve end-to-end prediction, we pad pixels around each image so their width and height are the multiples of 16 with reference to the network architecture (Supplementary Fig. 13). The padding pixels are the replications of the border pixels. Before prediction, the IMC images are normalized by the pre-calculated maximum of the corresponding channels in the training set. The outputs of the network are re-scaled and set as the denoised images. Given the trained denoising model, inference is fast. We are able to denoise IMC images with pixels of 1000 by 1000 less than 1 second per image on a single NVIDIA Quadro RTX 6000 GPU.

### Reference hot pixel removal and statistics-based shot noise filtering methods

### Semi-automated Ilastik-based background noise removal
The semi-automated strategy in[21] utilizes Ilastik segmentation[53] to remove background noise in IMC images. An expert annotates signal or background regions of IMC images, and then Ilastik trains a random forest classifier for background noise removal. To achieve good denoising quality, large areas of background require manual labeling, which is labor-intensive. Furthermore, low image quality may affect the accuracy of this method as well. After background removal, the images are binarized to solve batch effect issues. Then the single cell information is calculated by counting the positive signal frequency rather than the mean intensity of every single cell. Here, we only utilized Ilastik (version 1.3.2post1) for background noise removal of IMC images, and still applied the mean intensities as the single cell profiles. To better reveal the enhancement by DeepSNiF, we applied the same labels for the trainings of DIMR and DeepSNiF-processed images.

### MAUI
MAUI software package[7,19] includes spillover correction, noise removal and aggregate removal. All three steps require expert observation, which is also labor-intensive. Here, we only benchmarked the noise removal method in MAUI with our DeepSNiF algorithm. Briefly, it calculates the distances between a non-zero pixel and its K nearest non-zero neighbors, then builds a histogram based on the summations of the distances for all the non-zero pixels. Next, a threshold is manually selected to remove the pixels with larger summations by observing the distribution of the histogram. This method is based on the assumption that noisy regions look more sparse than normal regions. MAUI was implemented by the software package from https://github.com/angelolab/MAUI. The parameter K and the threshold were manually tuned to guarantee the best performance of MAUI (Supplementary Fig. 27).

### Pixel classification and cell segmentation
In single cell segmentation, the pixels in each image were defined as belonging to the nucleus, cytoplasm, or background compartment

using the pixel classification module of Ilastik[53] (version 1.3.2post1) as described in https://github.com/BodenmillerGroup/ImcSegmentation Pipeline. In our experiments, the DIMR and DeepSNiF-processed images were both set as inputs for cell mask segmentation. The DeepSNiF-based cell masks were primarily used for further analysis, while those based on DIMR were validated for robustness in some cases. The Random Forest classifier was trained on the channels including CD38, MPO, CD14, CD71, CD11b, CD4, CD20, CD8a, CD15, Ki-67, CD3, CD45RO CD235a, Histone-H3, and Iridium. This allowed for the generation of maps that integrate for each pixel the probability of belonging to each of three compartments. Based on the trained classifier, probability maps were generated for the whole dataset and exported as tiff files in batch mode.

Subsequently, CellProfiler[54] (version 3.1.8) was used to define cell masks for marker expression quantification. To define cell borders, nuclei were first identified as primary objects based on ilastik probability maps and expanded through the cytoplasm compartment until either a neighboring cell or the background compartment was reached. Cell masks were generated for identification of single cells and used to extract single-cell information from IMC images.

### Single-cell marker profile extraction and line fitting
We used HistoCAT[55] (version 1.7.6) to extract single-cell marker profiles based on the IMC images and their segmentation masks. All the data were not transformed and used directly.

We conducted bisqaure line fitting for the extracted DIMR and DeepSNiF-processed single cell data with customized MATLAB (R2021a, MathWork) scripts.

### Positive cell identification
For initial identification of marker-positive cells, we modified the method described in[14]. Briefly, univariate Gaussian mixture models with scikit-learn[56] (version 1.0.2) were used to estimate the positive thresholds of each marker. Before threshold estimation, all data were 99th-percentile normalized so that the impact of extremely bright cells can be eliminated.

For each channel, we performed model selection with models with 6–15 mixtures for DIMR data, in order to estimate the positive threshold accurately. We selected the model on the basis of the Davies-Bouldin index[57] and identified a positive threshold for a given channel by considering both the distributions of cell profiles and the overall IMC image intensities. The estimated positive thresholds of single cell data from DIMR and DeepSNiF-based cell segmentations are summarized in Supplementary Tables 12 and 13.

### Cell-type annotation
A subset of markers extracted from DeepSNiF-based cell segmentation masks of the human bone marrow dataset were utilized for cell phenotypic annotation, including CD38, MPO, CD14, CD71, CD11b, CD4, CD169, CD20, CD8a, CD15, CD3, and CD235a. Before analysis, data were 99th-percentile normalized followed by Z-score normalization. Then the DIMR data was clustered by the Phenograph algorithm with 20 nearest neighbors of each cell[46] with the Leiden community detection algorithm[47] with resolution of 6.0, which resulted in over clustering with 117 clusters. The generated clusters were manually labeled with a broad ontogeny and the channels that were most abundant in each cluster (Supplementary Fig. 35), resulting in 9 cell types, including immune cell subsets (B cell, CD4+ T cell, CD8+ T cell, and plasma cell, monocyte/macrophages), erythroid, myeloid, and other CD4+ cells and others.

The DeepSNiF data clustering and annotation utilized a weighted K-nearest neighbor (KNN) approach (K=20) to map the DeepSNiF data into the DIMR clusters. It first constructs a Jaccard graph between each cell from DeepSNiF and all the DIMR cells, and then maps the DeepSNiF data into the DIMR clusters with the shortest weighted distance. The

leave-one-out DeepSNiF data was also annotated with the same approach. The Phenograph with Leiden algorithms were implemented by the software packages from https://github.com/jacoblevine/PhenoGraph and https://github.com/vtraag/leidenalg. DeepSNiF data annotation was implemented with customized python scripts. The same markers extracted from DIMR-based cell masks were also analyzed with the same approach (Supplementary Fig. 38).

Notably, multiple strategies were applied to reduce the noise impact during DIMR clustering: (1) Z-score normalization is consistent for handling different sources of noise in multiplexed cell data, including low intensity signal, high background signal, segmentation noise, and imaging artifacts, as verified by[58]; (2) the Jaccard graph construction in Phenograph is robust to noise, which is verified in[46]; and (3) over-clustering could improve the clustering accuracy[58]. Besides, we didn't annotate the DeepSNiF and leave-one-out DeepSNiF data with the same approach of DIMR because (1) The community detection results by Leiden algorithm is random so that it is very difficult to compare the annotations from different data; and (2) the weighted KNN method for DeepSNiF and leave-one-out DeepSNiF clustering could clearly reveal the differences before and after the processing.

The manual cell-type annotation in Fig. 4 was based on the DeepSNiF-based cell segmentation masks. Briefly, DIMR and DeepSNiF images were overlaid with the cell masks in FIJI[59]. In some extremely noisy cases, the DIMR images were denoised by Gaussian filters to improve the annotation accuracy. Based on the signal in each cell mask, the cells were classified as B, CD8 T, CD4 T cells and other cells. Some positive signals were identified as hot clusters and discarded. The annotation results were manually recorded. To test the impact of segmentation masks on annotation results, we have also annotated the lymphocytes on DIMR-based cell masks (Supplementary Fig. 44).

### Fast interpolation-based *t*-SNE algorithm

For visualization, high-dimensional single-cell data of DIMR and DeepSNiF were reduced to two dimensions using the nonlinear dimensionality reduction algorithm fast interpolation-based *t*-SNE[48]. This algorithm was implemented by the software package in https://github.com/KlugerLab/FIt-SNE. Before the analysis, data were 99th-percentile normalized followed by Z-score normalization. The *t*-SNE parameters with perplexity of 50 and theta of 0.5 were used. The random seeds for the individual runs were recorded.

### Enrichment calculation of positive cell markers after DeepSNiF and leave-one-out DeepSNiF

To evaluate the effect of positive cell marker enrichment after DeepSNiF and leave-one-out DeepSNiF, the cell-type annotations before and after the processing were selected, and the percentage of positive markers on each cell types was calculated. The relative change was then defined as the difference between the percentage of positive markers after and before the processing.

### Enrichment calculation of cell types after DeepSNiF and leave-one-out DeepSNiF

To evaluate the effect of cell type enrichment after DeepSNiF and leave-one-out DeepSNiF, the positive cells for a given marker before and after the processing were selected, and the percentage of each cell type based on cell-type annotation was calculated. The relative change was then defined as the difference between the percentage of cell-type composition after and before the processing.

### Accuracy metrics

In simulation, the accuracy metrics including root mean squared error, peak SNR and structural similarity[45] are used to access the image qualities because of the availability of ground truths. They are defined in Supplementary Note 3.2 in detail.

For the real experimental data, five types of metrics were used for quantitative evaluations. The standard deviation of background (STDB) and contrast-to-noise ratio (CNR) were used to evaluate the noise level and contrast of IMC images. CNR is defined as Eq. (3),

$$CNR = (C_{sig} - C_{bg})/\sigma_{bg} \tag{3}$$

where $C_{sig}$ and $C_{bg}$ are the mean of the signal and background and $\sigma_{bg}$ is the STDB. In this metric, the signal and background regions of IMC images are manually annotated.

Pearson's correlation coefficient (PCC) was used as the metric to reflect the similarity between two groups of data. The PCC between measured data $\mathbf{Y}$ and the reference $\mathbf{Y}_{ref}$ is defined as Eq. (4),

$$PCC(\mathbf{Y},\mathbf{Y}_{ref}) = \frac{E[(\mathbf{Y} - \mu_{\mathbf{Y}})(\mathbf{Y}_{ref} - \mu_{\mathbf{Y}_{ref}})]}{\sigma_{\mathbf{Y}}\sigma_{\mathbf{Y}_{ref}}}, \tag{4}$$

where $\mu_{\mathbf{Y}}$ and $\mu_{\mathbf{Y}_{ref}}$ are the mean values of images $\mathbf{Y}$ and $\mathbf{Y}_{ref}$, respectively; $\sigma_{\mathbf{Y}}$ and $\sigma_{\mathbf{Y}_{ref}}$ are the standard deviations of $\mathbf{Y}$ and $\mathbf{Y}_{ref}$, respectively; and $E$ represents arithmetic mean.

Furthermore, F1 score was used to evaluate the accuracy of background noise removal. F1 score and Jaccard score were used to evaluate the accuracy of cell annotation of B, CD8 T and CD4 T cells, which can be formulated as Eqs. (5) and (6), respectively.

$$F1\ score = \frac{2TP}{2TP + FP + FN}, \tag{5}$$

$$Jaccard\ score = \frac{TP}{TP + FP + FN}, \tag{6}$$

where TP, FP and FN are the pixel number of true positives, false positives and false negatives, respectively. All of the evaluation process was implemented with customized MATLAB (R2021a, MathWork) scripts. RMSE, PSNR, SSIM, PCC, F1 score, and Jaccard score were computed using MATLAB built-in functions.

### Statistical analysis

Other than specially stated, quantitative data are presented as box-and-whisker plots (center line, median; limits, 75% and 25%; whiskers, maximum and minimum). The two-sided Wilcoxon matched-paired test was used for the statistical significance determination of repeated measurements. The two-sided Kolmogorov-Smirnov test was used for the statistical significance determination of different distributions in Fig. 4d. All the statistical tests are implemented with Prism 9 (Graph-Pad Software Inc.). Statistical significance at $P < 0.05$, 0.01, 0.001, and 0.0001 are denoted by *, **, *** and ****, respectively. "ns" means "no significance".

### Reporting summary

Further information on research design is available in the Nature Portfolio Reporting Summary linked to this article.

## Data availability

The human bone marrow IMC data and simulated data generated in this study are available from Zenodo https://doi.org/10.5281/zenodo.6533905[60]. The human pancreatic cancer IMC dataset[10] can be downloaded from https://hpap.pmacs.upenn.edu/, the human breast cancer IMC dataset[12] can be downloaded from https://doi.org/10.5281/zenodo.3518284, and the MIBI data set[19] can be downloaded from https://github.com/angelolab/MAUI, respectively. Source data for Figs. 1b, e, g, 2c, 3b, e, f, 4b, d–g, and Supplementary Figs. 2, 3–11a, 18–20b, 25b, 26b, 28–30b, 32b, 38–43, and 44b are provided in the

source data file. All the other data supporting the results in this paper can be accessed from https://doi.org/10.5281/zenodo.7336448[61]. Source data are provided with this paper.

## Code availability

The code used in this study and the corresponding tutorial are publicly available at GitHub https://github.com/PENGLU-WashU/IMC_Denoise and Zenodo https://doi.org/10.5281/zenodo.7559428[62].

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

## Acknowledgements

Research reported in this publication was supported in part by NIH NCI R01CA240711, R01CA229893, K12 CA167540, and 1P50CA171963; NHLBI R21HL150636; American Society of Hematology Scholar Award; and Evans Foundation Edward P. Evans Center for MDS. We thank the Alvin J. Siteman Cancer Center at Washington University School of Medicine and Barnes-Jewish Hospital in St. Louis, MO., for the use of the Bursky Center for Human Immunology and Immunotherapy Programs Immunomonitoring Laboratory, which provided IMC service. The Siteman Cancer Center is supported in part by NCI Cancer Center Support Grant #P30 CA091842. The content is solely the responsibility of the authors and does not necessarily represent the official views of the NIH.

## Author contributions

P.L., K.A.O., and D.L.J.T. conceived and designed the project. P.L. developed and implemented the software. K.A.O., D.E.B., S.T.O., M.B.R., D.A.C.F., and D.C.L. obtained the human bone marrow tissues and performed the IMC staining and imaging. P.L., K.A.O., and D.L.J.T. conducted image and downstream analysis. R.K.P., W.N.B., K.G.S., S.T.O., and D.C.L. helped with analysis. P.L. and K.A.O. conducted the manual annotation with the further verification from all authors. D.L.J.T. supervised the project. P.L., K.A.O., and D.L.J.T. wrote the manuscript with input from all authors.

## Competing interests

The authors declare no competing interests.

## Additional information

[1]Department of Biomedical Engineering, Washington University in St. Louis, St. Louis, USA. [2]Department of Radiology, Mallinckrodt Institute of Radiology, Washington University School of Medicine, St. Louis, USA. [3]Program in Quantitative Molecular Therapeutics, Washington University School of Medicine, St. Louis, USA. [4]Department of Medicine, Washington University School of Medicine, St. Louis, USA. [5]The Bursky Center for Human Immunology and Immunotherapy Programs Immunomonitoring Laboratory, Washington University School of Medicine, St. Louis, USA. [6]Department of Pathology & Immunology, Washington University School of Medicine, St. Louis, USA. [7]Department of Oncology, Sidney Kimmel Comprehensive Cancer Center (SKCCC), Johns Hopkins University, Baltimore, USA. [8]Department of Urology, James Buchanan Brady Urological Institute, Johns Hopkins University School of Medicine, Baltimore, USA. [9]Department of Radiation Oncology, Washington University School of Medicine, St. Louis, USA. [10]Oncologic Imaging Program, Siteman Cancer Center, Washington University School of Medicine, St. Louis, USA. [11]These authors contributed equally: Peng Lu, Karolyn A. Oetjen. ✉e-mail: thorekd@wustl.edu

