## [Peer Review File · Nature Communications]

Reviewers' Comments:

Reviewer #1:

Remarks to the Author:

IMC-Denoise is a new approach that addresses a common problem with Imaging Mass Cytometry data: random hot pixels and shot noise background. Most commonly used analysis pipelines only use fairly basic filtering strategies, and usually accept a significant residual amount of these two noise factors. A simple all-round algorithm to improve noise reduction would certainly contribute to the field.

While I do not have the expertise to assess the soundness of the mathematical formulations or algorithms used, the presented images look very clean and sharp. However, I have a few concerns related to the fairness of the comparisons made prove superiority to other methods in some of the figures, that I have set out below.

Figure 2a: segmentation based on thresholding. Reference 21 suggests to find a threshold for every marker and every image individually to identify background signal. It looks like in the example shown here (DIMR_thresh) this manual threshold was not chosen at the best possible intensity level, including a lot of background. This gives an unfair advantage to other methods presented (and unfair discredit to the ref 21 Ijsselsteijn paper).

Supp figs 27/28 & fig 2e-h: compares single cell profiles from the different noise removal approaches, all using the cell mask from DeepSNF. I appreciate that it is preferred to minimise variability and stick with one cell mask. But, is DeepSNF not always favoured in the results if the cell mask used was based on DeepSNF? (Taken to the extreme: if the raw data projects circles, but DeepSNF gives only triangles as output, then a triangular mask will favour DeepSNF in every downstream analysis).

Perhaps the reciprocal experiment might be done as well: comparing raw, DIMR and DeepSNF using a single cell mask based on raw or DIMR data?

It is not clear what cell mask was used to produce figure 3. Again only DeepSNF, or each their own?

Figure 3c shows how cell annotation when using either DIMR or DeepSNV: These figures are very difficult to interpret, a higher magnification would help. Figure 3d is a better example.

Figure 3d and e: When looking at the image in close-up, it becomes apparent that the data in DeepSNF looks very far from the original data. Much more seems to have changed than just filtering out background noise. Negative pixels seem filled in as positive, the image looks "enhanced". When calculating the positive marker percentage in figure 3d, is that using these "enhanced" values, and how real are these? If the signal per cell has been enhanced, then it is of no surprise that the percentage positive marker per cell type is higher.

And following from that: Is the mean signal per individual cell still the same and is the signal still linear? Can the DeepSNV data be used to say something about expression levels between cells?

Figure 4a: It is not clear what the manual annotations were based on. Drawing of cell outlines on raw data images? Or just cell type annotations using the same cell mask? Who performed the manual annotations?

Overall assessment: I think the results look beautiful and the clean-up is impressive, but I have some doubt whether the comparisons made are often favouring the DeepSNV outcome. And besides showing the improvement, I'd like to also see that the signal is still true to the original, not just a neural network's interpretation. Can the DeepSNV data faithfully be used to say something about expression levels between cells? Or has the underlying data been altered too much?

Finally, IMC analysis is very complicated and this is a barrier for many users. A simple all-round algorithm to improve noise reduction would certainly contribute to the field. While IMC-denoise potentially improves the quality of the data significantly, it has not been made clear how others might make use of this technology. How simple would it be to run for a new user? Are there any

input variables that need to be optimised? How much programming skill would be required? I miss details on implementation, especially as the authors end with: "We expect IMC-Denoise to become a widely used pipeline in IMC analysis due to its adaptability, effectiveness and flexibility". A schematic figure describing the workflow for a user of IMC-Denoise would be helpful.

Minor comments:

DeepSNF as a name for the algorithm is not original and may cause confusion. SNF has been used as abbreviation for Similarity Network Fusion and based on this there is even a paper describing DeepSNF:

Luciano & Hamza, 2019

<https://doi.org/10.1007/s00371-019-01668-9>

The introduction mentions decalcification required bone marrow samples, but the methods section does not mention whether this was applied to the tissues analysed here.

The results section compares many different alternative methods for filtering noise. The many abbreviations for these methods makes the text quite difficult to read for those that are not familiar with these methods.

The last paragraph discussing fig 2b and 2c should be revised, due to repetition and some inconsistency (p10).

Labelling of figure 4a, should CD20 be green in the left panels?

Reviewer #2:

Remarks to the Author:

What are the noteworthy results?

- Lu et al. describe a novel denoising pipeline for highly multiplexed IMC images which involves two steps: differential intensity map-based restoration (DIMR) and a self-supervised deep learning algorithm for filtering shot noise (DeepSNF). Lu et al. demonstrate the use of this pipeline for enhancing the cell intensities of 12 markers for 96232 cells that they claim resulted in analysis similar to manual annotation of each dataset

Will the work be of significance to the field and related fields? How does it compare to the established literature? If the work is not original, please provide relevant references.

- This work is relevant to all fields which make use of histopathology and immunohistochemistry techniques, including but not limited to cancer and autoimmunity.
- Current methods of removing image noise are imperfect and improvements are needed.
- IMC allows to visualize and extract data from >50 analytes on a single tissue section. This is of value because the interpretation of some analytes is only possible in the context of other analytes; because some samples are rare and irreplaceable, because it can be difficult to visualize low density antigens against background using standard immunohistochemistry methods such as immunofluorescence.

Does the work support the conclusions and claims, or is additional evidence needed?

- Yes the work supports the claims and conclusions. Denoising with DIMR and DeepSNF appear to reduce background to a comparable degree or in some cases better than other algorithms used in the literature, but also appears to reduce resolution visually.
- Fig 1f: some markers seem to disappear after DeepSNF denoising (ie CD31, CD20). Is this biologically true? Were there other analytes tested in the same tissue to corroborate the results (ie CD19 for B cells?)
- Fig 2h is showing that CD20+ gated cells express some amount of MPO. Is this a biological truth or is it an artifact of the image segmentation? Same with CD3+ and CD20+ gated populations having some expression of CD11b

Are there any flaws in the data analysis, interpretation and conclusions? - Do these prohibit publication or require revision?

- With the disclosure that I am not an expert on computer algorithms and bioinformatic analysis, it does seem that the data analysis and interpretation support the conclusion, with the caveats stated above

Is the methodology sound? Does the work meet the expected standards in your field?

- Yes, the initial segmentation analysis is based on a well-established pipeline published by the Bodenmiller group

Is there enough detail provided in the methods for the work to be reproduced?

- Yes

Additional comments/questions:

Authors are encouraged to discuss the questions posed below:

- What is the maximum amount of "noise" before a channel is deemed unable to denoise? Is there an empirical way to determine this?

- Have analytes that have low signal-to-noise ratio been tested?

- How would the pipeline adapt to analytes that should visualize targets localized in-between nucleated cells (ie. synapses)? Denoising these channels often results in reduction of image intensity in areas where a signal is expected.

- How would the software handle instances where cell signal is unexpected; ie if a novel cell type is discovered, how would they distinguish between true positive and false positive? Has this been tested on tissues where there is a mix of morphologically heterogenous cells? (ie brain tissue where there are a mix of glial cells, immune cells, and fibroblasts). What happens when there are processes extending from glial cells that are present in the section but their corresponding nuclei are not in the same plane?

Reviewer #3:

Remarks to the Author:

This study develops IMC-Denoise, a denoising automated pipeline to enhance Imaging Mass Cytometry (IMC) images. Two primary sources of noise are here addressed: hot pixels and shot noise. To remove hot pixels, IMC implements a differential intensity map-based restoration (DIMR); to filter shot noise, it implements a deep learning method, DeepSNF. The approach presented here is of intrinsic interest as it develops some new algorithms and approaches that are illuminating and useful. The extensive benchmarking and comparisons with existing denoising methods are strengths of the study. The end-to-end analysis, including the impact on automated cell phenotyping, is important and further strengthens the study. Some concerns that should be addressed to improve the interpretability and, hopefully, the utility of the study are listed below.

1. How many iterations of DIMR does it take to adequately remove the hot pixels in data? Some quantification of the variability would be very helpful. (The paper indicates $n = 4$ and iteration number is set at 3.) How computationally intensive is the approach?

2. The definition of the threshold point x_T (top of page 6) as the value of x at which the second derivative, evaluated at $x-dx$, of the fitted curve from the kernel density estimation algorithm is greater than or equal to 0 (while the second derivative evaluated at x is less than or equal to zero) is mathematically imprecise and problematic. The " $x-dx$ " bit is problematic. Please clarify this definition.

3. In the expression for the loss function of the DeepSNF model: The pixel mask is missing from equations 2 (and equation 31) in the regularization term (with coefficient given by the Hessian norm regularizer).

4. It would be useful for the authors to provide a justification for the specific choice of neural

network architecture. The deep learning methodology is given in very general terms and is somewhat of a blackbox.

The manuscript can also benefit from thorough language editing.

Other

1. In description of equation 2 in the paragraph immediately below, shouldn't the M be replaced by M_p (i.e. the pixel mask). M is not a separate variable in equation 2.

Response to Reviewers:

Overview:

The revised manuscript and supplementary materials have been uploaded for further review. The detailed responses to the reviewer comments are below, in the order received. We have enumerated and formatted our responses (in italicized *blue text*) to ensure that it can be clearly followed. Data and figure elements are enclosed throughout, in order to assist the Reviewers to follow the rationale for our responses and assess improvements. Where these changes have been made in the main manuscript and the supplemental files are listed.

Before the point-by-point response, we have a few general notes to the Reviewers:

- Per a comment from Reviewer 1, we have renamed DeepSNF to DeepSNiF to avoid confusion with other algorithms. All of the references to the algorithm in this response letter and throughout the manuscript have been amended.
- The initial submission of the manuscript included rigorous evaluation and comparison with both conventional and state of the art approaches to denoising and classification tasks in highly multiplexed single cell data. In the resubmission, we further expand the scope of these comparisons and the basis for our parameter choices. These additional experiments, performed on both synthetic and real-world data, underline the utility and wide applicability of DeepSNiF and IMC-Denoise.
- We include amended figures directly in this response letter to ease the Reviewer evaluation of our responses.

Reviewer #1:

R1.0: IMC-Denoise is a new approach that addresses a common problem with Imaging Mass Cytometry data: random hot pixels and shot noise background. Most commonly used analysis pipelines only use fairly basic filtering strategies, and usually accept a significant residual amount of these two noise factors. A simple all-round algorithm to improve noise reduction would certainly contribute to the field. While I do not have the expertise to assess the soundness of the mathematical formulations or algorithms used, the presented images look very clean and sharp. However, I have a few concerns related to the fairness of the comparisons made prove superiority to other methods in some of the figures, that I have set out below.

R1.1: Figure 2a: segmentation based on thresholding. Reference 21 suggests to find a threshold for every marker and every image individually to identify background signal. It looks like in the example shown here (DIMR_thresh) this manual threshold was not chosen at the best possible intensity level, including a lot of background. This gives an unfair advantage to other methods presented (and unfair discredit to the ref 21 Ijsselsteijn paper).

We certainly were not intending to draw any unfair comparisons in our work. As an example, when comparing DIMR to NTHM and MTHM hot pixel removal methods, we performed this on the basis of optimal parameters for these other methods. To the present comment regarding thresholding, we have undertaken further evaluation to provide a comprehensive and fair comparison. In order to ensure that manual thresholds chosen are optimal for the comparisons in benchmarking, we include multiple thresholds for all the CD34 and Collagen III images used in Figure 2a-c. For DIMR data, we have tested thresholds 1-4 for each marker and each image to select the best background removal results. The thresholds are all integers because the pixel values of raw and DIMR-processed images are all integers. We

then select the best results for each image from all the test cases. We have added the following figure as new Supplementary Figure 25 on Page 46 of the Supplementary Materials, and discuss this approach in Results at lines 247-251. This result shows different threshold values and the impact on correctly or incorrectly identifying molecularly specific stained structures, and the attendant F1 scores.

New Supplementary Fig. 25 Optimal threshold selection for DIMR_thresh on background removal. Because the pixel values are non-negative integers in DIMR-processed images, threshold values from 1 to 4 are selected to remove background noise of DIMR-processed CD34 and Collagen III images, and sub panel (a) and (b) represent the visual comparison and quantitative evaluations results, respectively. The optimal thresholds are then chosen for DIMR_thresh (CD34: 2; Collagen III: 1) to compare with other methods in Fig. 2b–c. Scale bar: top: 50 μ m, bottom: 75 μ m.

For DeepSNiF-processed images, we also selected multiple thresholds from 0.9 to 1.2 because the pixel values are continuous. Note that in this case, a single threshold was selected for all the images per marker for DeepSNiF-processed images, because we find this has already achieved good background removal performance (an advantage of our method). We have added the following figure as new Supplementary Figure 26 on Page 47 of the Supplementary Materials.

New Supplementary Fig. 26 Optimal threshold selection for DeepSNiF_thresh method on background removal. Threshold values from 0.9 to 1.2 are selected to remove background noise of DeepSNiF-processed CD34 and Collagen III images, and sub panel (a) and (b) represent the visual comparison and quantitative evaluations results, respectively. The optimal thresholds are then chosen for DeepSNiF_thresh (CD34: 1; Collagen III: 1) to compare with other methods in Fig. 2b–c. Scale bar: top: 50 μm , bottom: 75 μm .

Using these more thorough comparisons, Fig. 2a-c have been also revised. We now use representative panels of optimally selected threshold values. In addition, we have made some edits to clarify this modification on Line 247-251 on Page 10 of the article file (see redline version).

New Fig.2a-c IMC-Denoise enables background noise removal and enhances downstream analysis of the human bone marrow IMC dataset. (a) Examples of DIMR and DeepSNiF-processed IMC images labeled with CD34 and Collagen III. (b) Visual inspection of background removal results of DIMR and DeepSNiF-processed images, in which DIMR_thresh and DeepSNiF_thresh are binarized with the optimal thresholds (Supplementary Figs. 25 and 26), DIMR_IIastik and DeepSNiF_IIastik are segmented by the IIastik software package, and MAUI results are the DIMR images processed by the MAUI software package (Supplementary Fig. 27), respectively. Manual annotated images are served as ground truths. (c) After DeepSNiF denoising, the background removal accuracy improves significantly in terms of F1 score, for both CD34 ($n=15$) and Collagen III-labeled images ($n=12$). Notably, DeepSNiF_IIastik achieves the highest accuracy, while DeepSNiF_thresh performs better than all the background removal results from DIMR images.

To summarize: The conclusions from these updated results remain **consistent with the initial manuscript**; specifically that DeepSNiF provides a means to generate background removal and denoising with minimal user-variable interaction. Manual threshold value choice has an impact on the results of the comparison with existing methods, but the magnitudes of these changes are small; DeepSNiF-based methods achieve best-in-class results even though the optimal thresholds were selected for DIMR_thresh.

R1.2: Supp figs 27/28 & fig 2e-h: compares single cell profiles from the different noise removal approaches, all using the cell mask from DeepSNiF. I appreciate that it is preferred to minimise variability and stick with one cell mask. But, is DeepSNiF not always favoured in the results if the cell mask used was based on DeepSNiF? (Taken to the extreme: if the raw data projects circles, but DeepSNiF gives only triangles as output, then a triangular mask will favour DeepSNiF in every downstream analysis). Perhaps the reciprocal experiment might be done as well: comparing raw, DIMR and DeepSNiF using a single cell mask based on raw or DIMR data?

The Reviewer is correct – we had used a single mask as a means to provide a direct comparison across methods. We thank the reviewer for the suggestion of evaluating multiple masks to further assess performance of existing approaches and our novel method – and have performed the Reviewer suggested experiment.

To do so, we have extracted the raw, DIMR, DIMR_IIastik and DeepSNiF single cell data from the segmentation masks of DIMR images. We have added the single cell data profile comparisons as new Supplementary Fig. 33 on Page 54 of the Supplementary Materials. These plots show that the characteristics of the single cell data from DIMR-based segmentation masks is similar to that from DeepSNiF-based masks (Supplementary Figs. 31 and 32b on Pages 52 and 53 of the Supplementary Materials).

New Supplementary Fig. 33 The impact of DIMR on single cell data extracted from DIMR-based cell segmentation masks. (a) Each sub-figure represents the one-on-one relationship between the raw and DIMR data of a particular marker in single cell scale. The bottom right value in each sub-figure represents the percentage of the difference between the raw and DIMR data. (b) Each sub-figure represents the one-on-one relationship between the DIMR and DeepSNiF data of a particular marker in single cell scale. The bottom right value in each sub-figure represents the slope of the line fitting results and the PCC between the DIMR and DeepSNiF data. These values indicate the DIMR and DeepSNiF single cell data are at the same scale and linearly correlated.

The impact of such different masks on cell phenotyping was also evaluated. We have added the new manual gating results as new Supplementary Fig. 34 on Page 55 of the Supplementary Materials.

New Supplementary Fig. 34 Evaluations of denoising algorithms with manual gating strategies on single cell data extracted from DIMR-based cell segmentation masks. The numbers in these panels are the cell percentages of the corresponding ranges. DIMR slightly enhances the single cell analysis over raw data, while DeepSNiF further enhances the DIMR results and performs better than semi-automated DIMR_lassikprocessing.

Both the results in new Supplementary Figs. 33 and 34 are similar to the results from DeepSNiF-based segmentation masks. This indicates that our denoising algorithm is robust to different single cell segmentation masks. Thus, we may conclude that that this is not a circles-to-triangles comparison. We have added a new paragraph to describe the new results on Line 294-298 from Pages 11 to 12 of the article file (see redline version).

R1.3: It is not clear what cell mask was used to produce figure 3. Again only DeepSNF, or each their own?

In the original version of this manuscript, the DeepSNiF-based cell masks were used to generate the results in Fig. 3. We regret that this was not more clear. In this revised manuscript, we have added the automated cell phenotyping analysis with the DIMR-based cell masks as (new) Supplementary Figure 38 on Page 59 of the Supplementary Materials. We find the results are similar to those generated by DeepSNiF-based cell masks. Regardless of the cell mask that was employed, DeepSNiF can remarkably enhance the phenotyping results. We have added additional discussion for these new results on Line 344-347 on Page 13 of the article file (see redline version).

New Supplementary Fig. 38 DeepSNiF enhances automated cell phenotyping on human bone marrow IMC data with DIMR-based cell segmentation masks. (a) t-SNE plots of DIMR and DeepSNiF with cell phenotyping results. (b) The relative change in cell phenotypes before and after DeepSNiF. (c) DeepSNiF enhances the sensitivity of cell phenotyping. After DeepSNiF processing, the non-specific marker signals reduce while the specific ones enrich in the cell types, respectively. The circle size indicates the positive marker percentage in a particular phenotype of DIMR, and the circle colour indicates the relative changes of the positive rate for the particular markers after DeepSNiF enhancement. (d) DeepSNiF enhances the specificity of cell phenotyping. With DeepSNiF denoising, the ratios of specific phenotypes increase while those of non-specific phenotypes decrease in the positive markers. The relative change is the difference in percentage composition of each cell type before and after DeepSNiF enhancement.

R1.4: Figure 3c shows how cell annotation when using either DIMR or DeepSNiF: These figures are very difficult to interpret, a higher magnification would help. Figure 3d is a better example.

We thank the Reviewer for this comment. Demonstrating the impact of our approach to improve the image quality and quantitation is a challenge, and we appreciate the suggestion to reduce the field of view. We have revised Fig. 3c with higher magnification images. We have also provided a new Supplementary Fig. 36 on Page 57 of the Supplementary Materials.

New Fig. 3c Comparisons of DIMR and DeepSNiF-processed IMC images labeled with different cell markers, and the corresponding cell annotation results. The sub-panels (i)–(iv) in (c) correspond to the white dashed box region selection in their first panels, respectively. The white contours represent the differential phenotyping results between DIMR and DeepSNiF.

A lower magnification (wider field of view) image is included in New Supplementary Fig. 36 Comparisons of DIMR and DeepSNiF-processed IMC images labeled with different cell markers, and the corresponding cell annotation results with the DeepSNiF-based cell segmentation masks (Fig. 3(c)). The bottom row corresponds to the white dashed box regions in the top row images. The white contours represent the differential phenotyping results between DIMR and DeepSNiF. Scale bar: Top: 145 μm , bottom: 50 μm .

R1.5: Figure 3d and e: When looking at the image in close-up, it becomes apparent that the data in DeepSNiF looks very far from the original data. Much more seems to have changed than just filtering out background noise. Negative pixels seem filled in as positive, the image looks “enhanced”. When calculating the positive marker percentage in figure 3d, is that using these

“enhanced” values, and how real are these? If the signal per cell has been enhanced, then it is of no surprise that the percentage positive marker per cell type is higher.

The Reviewer asks why the data from the denoised images looks different. First, we will mention that in response to the comment above (R1.4), we have revised this figure to show the images in greater magnification to facilitate easier interpretation by the reader. Additionally, we have modified the scale of circle diameter in 3E to more clearly demonstrate the relative positive marker change. Figure 3D shows the DIMR and DeepSNiF-processed IMC images of bone marrow with CD3, CD4 and CD8a specific markers, and the corresponding binary cell annotation results. In the next panel, Figure 3E illustrates the positive marker percentage across 12 select immune targets, and relative sensitivity of improvements by DeepSNiF. The Reviewer notes that the images (Fig.3D) between DIMR (hot pixel removed) and DeepSNiF-denoised data appear different. This is correct. The reason that the data in DeepSNiF appear different is that the original data in CD3, CD4 and CD8a channels are extremely noisy. These are widely recognized in the field as difficult-to-stain markers because of background signal; these effects are only increased in the complex and further difficult-to-stain marrow. With DeepSNiF, a high amount of background noise has been filtered. Note that there exists noise in signal as well, under these low SNR conditions. This could be the features to which the Reviewer remarks as looking like noise from negative pixels.

In the further two right panels of Fig. 3C, white contours were applied to the cell outlines for which DeepSNiF changes the cell annotation result. As an example: a cell mask filled in as CD8 (red) in the rightmost (DeepSNiF annotation) panel that is empty (grey) in the DIMR annotation shows how denoising can improve the cell-calls from low SNR data.

The denoising effects on the image are not a result of efforts to “enhance” our image pixel values nor single-cell signal. Several analyses were included in the original manuscript, and expanded here, to prove this statement. First, we have calculated peak SNR (PSNR) and structural similarity (SSIM) in our simulation experiments (revised Supplementary Figs. 8-11 from Page 28 to 31 of the Supplementary Materials). These results demonstrate that the PSNR and SSIM improve after denoising, which means the pixel values after denoising are closer to the ground truth values. Second, we have conducted line fitting for DIMR and DeepSNiF-processed single cell data (revised Supplementary Figs. 32a and 33b on Page 53 to 54 of the Supplementary Materials and shown below). This analysis shows that the range of DIMR and DeepSNiF single-cell data are highly similar (correlated), no matter which cell segmentation masks are used. The slopes of the fitted line are almost 1 for all the markers. Across all the markers, the slope of this correlation is nominally 1, except for the CD20 marker. The reason for the exclusive result from CD20 is because of the very high degree of noise in this channel. The large amount of low intensity noise is almost filtered towards 0, which may bias the line fitting. Nevertheless, the ranges of the DIMR and DeepSNiF-processed data are very close. To summarize, we have not “enhanced” our data. Rather, the higher percentage positive marker results from the decreased false positive due to improved image and data quality following denoising.

New Supplementary Fig. 32a Each sub-figure represents the one-on-one relationship between the DIMR and DeepSNiF data of a particular marker in single cell scale. The bottom right value in each sub-figure represents the slope of the line fitting results and the PCC between the DIMR and DeepSNiF data. These values indicate the DIMR and DeepSNiF single cell data are at the same scale and linearly correlated.

New Supplementary Fig. 33b Each sub-figure represents the one-on-one relationship between the DIMR and DeepSNiF data of a particular marker in single cell scale. The bottom right value in each sub-figure represents the slope of the line fitting results and the PCC between the DIMR and DeepSNiF data. These values indicate the DIMR and DeepSNiF single cell data are at the same scale and linearly correlated.

R1.6: And following from that: Is the mean signal per individual cell still the same and is the signal still linear? Can the DeepSNV data be used to say something about expression levels between cells?

The Reviewer asks if there is a change in signal intensity of markers on a single-cell basis following denoising. Within the revised Supplementary Figs. 32a and 33b from Page 53 to 54 of the Supplementary Materials and shown above (in response to R1.5), we have conducted line fittings and calculated Pearson correlation coefficients for all the markers that we have used in downstream analysis. The results demonstrate the mean signal per individual cell is the same and linear. DeepSNiF does not intend nor result in changes to the scale or linearity of the original cells' marker intensity. Instead, the approach works by correcting the bias resulting from noise in the image formation process. Furthermore, as shown with new Supplementary Fig. 32b on Page 53 and Supplementary Note 1.2.1 on Page 4 of the Supplementary Materials, lower signal is associated with higher shot noise. The implication for this is that more low-quality data will be restored by DeepSNiF.

R1.7: Figure 4a: It is not clear what the manual annotations were based on. Drawing of cell

outlines on raw data images? Or just cell type annotations using the same cell mask? Who performed the manual annotations?

This is an important question. The original manual annotations were based on the DeepSNiF-based cell masks. Previously, we have drawn the outlines for comparisons (Fig. 2a-c), which is based on pixel intensities. However, for cell scale comparison, the mask-based annotation method is preferred in order to directly compare the automated phenotyping results of DIMR and DeepSNiF. The manual annotations and verifications were performed by contributing authors. We have added a new paragraph to describe the new results on Line 682-688 on Page 26 and Line 757-758 on Page 30 of the article file (see redline version). We have also compared the annotation results on the DIMR-based cell masks (new Supplementary Fig. 44 on Page 65 of the Supplementary Materials). While there are some variations due to the differences from segmentation masks, this comparison demonstrates accuracy improvements as well.

New Supplementary Fig. 44 DeepSNiF enhances lymphocyte analysis. (a) Manual annotations for lymphocytes and comparisons with DIMR and DeepSNiF phenotyping results with DIMR-based cell masks. The white contours represent the differential phenotyping results between the annotated and DIMR/DeepSNiF results. (b) Annotation evaluations of DIMR and DeepSNiF by both Jaccard and F1 scores.

R1.8: Overall assessment: I think the results look beautiful and the clean-up is impressive, but I have some doubt whether the comparisons made are often favouring the DeepSNV outcome. And besides showing the improvement, I'd like to also see that the signal is still true to the original, not just a neural network's interpretation. Can the DeepSNV data faithfully be used to say something about expression levels between cells? Or has the underlying data been altered too much?

We thank the Reviewer for their positive interpretation of this work. In particular we are encouraged by the Reviewer comment about image quality – something that is hard to measure with real-world data – but certainly improves our ability to assess and navigate these complex datasets. Further, we have endeavored to show that there are objective improvements in the image and single-cell data quality following denoising, and that these results exceed conventional approaches through comprehensive comparison.

Through clarification and additional experimentation and analysis, we have further validated that we have not changed the scale and linearity of the single-cell data. The DeepSNiF algorithm filtered the noise in the image so that image quality and downstream analysis results have been improved. We also included comparisons to paired immunofluorescence images for additional reassurance that DeepSNiF is true to the original image signal (Supp. Fig. 23).

R1.9: Finally, IMC analysis is very complicated and this is a barrier for many users. A simple all-round algorithm to improve noise reduction would certainly contribute to the field. While IMC-denoise potentially improves the quality of the data significantly, it has not been made clear how others might make use of this technology. How simple would it be to run for a new user? Are there any input variables that need to be optimised? How much programming skill would be required? I miss details on implementation, especially as the authors end with: “We expect IMC-Denoise to become a widely used pipeline in IMC analysis due to its adaptability, effectiveness and flexibility”. A schematic figure describing the workflow for a user of IMC-Denoise would be helpful.

We thank the Reviewer for this comment. Previously, we had uploaded our software package on our GitHub page (https://github.com/PENGLU-WashU/IMC_Denoise). We have provided a detailed tutorial file that covers the installation and implementation of the software package. In order to ensure that the community is able to access and implement our approach and findings, we have added to this revision a schematic figure (revised Supplementary Fig. 12) and described this figure in new Supplementary Note 4 (Page 32-33 on the Supplementary Materials) that describes each individual step of the process. We have provided multiple tutorials in the Jupyter Notebook for those users with limited programming experiences. For those with some coding skills, scripts files with parameter descriptions have also been provided. We hope our tutorials and descriptions can be used to help users use our software package more easily. The GitHub page is kept updated, and users can post their questions on the page, so that we can help them in their specific cases.

Minor comments:

R1.10: DeepSNF as a name for the algorithm is not original and may cause confusion. SNF has been used as abbreviation for Similarity Network Fusion and based on this there is even a paper describing DeepSNF: Luciano & Hamza, 2019 <https://doi.org/10.1007/s00371-019-01668-9>

We were not aware, and thank the Reviewer for this information. We have changed DeepSNF to DeepSNiF (Deep learning-based Shot Noise image Filtering) in the manuscript, figures, supplemental materials, and software package. Further we have confirmed that there are no other algorithms with the same name.

R1.11: The introduction mentions decalcification required bone marrow samples, but the methods section does not mention whether this was applied to the tissues analysed here.

The decalcification step was applied to our tissues. We have added the descriptions from Line 512 to 513 in the article file (redline version).

R1.12: The results section compares many different alternative methods for filtering noise. The many abbreviations for these methods make the text quite difficult to read for those that are not familiar with these methods.

We appreciate this feedback. We have added a summary table for these algorithms in Methods section on Line 601. Furthermore, the detailed descriptions of these algorithms have been discussed in Supplementary Note 2.

R1.13: The last paragraph discussing fig 2b and 2c should be revised, due to repetition and some inconsistency (p10).

The text has been revised in the results and discussion for Fig 2.

R1.14: Labelling of figure 4a, should CD20 be green in the left panels?

There are multiple colors used in Fig. 4a. Therefore, color crosstalk is easily generated. To make the figure clearer, we label CD20 with gray in the revised figure. As a result, B cells are white in the figure. The figure legend has been amended as well.

Reviewer #2:

R2.1: What are the noteworthy results? - Lu et al. describe a novel denoising pipeline for highly multiplexed IMC images which involves two steps: differential intensity map-based restoration (DIMR) and a self-supervised deep learning algorithm for filtering shot noise (DeepSNF). Lu et al. demonstrate the use of this pipeline for enhancing the cell intensities of 12 markers for 96232 cells that they claim resulted in analysis similar to manual annotation of each dataset

Will the work be of significance to the field and related fields? How does it compare to the established literature? If the work is not original, please provide relevant references.

- This work is relevant to all fields which make use of histopathology and immunohistochemistry techniques, including but not limited to cancer and autoimmunity.
- Current methods of removing image noise are imperfect and improvements are needed.
- IMC allows to visualize and extract data from >50 analytes on a single tissue section. This is of value because the interpretation of some analytes is only possible in the context of other analytes; because some samples are rare and irreplaceable, because it can be difficult to visualize low density antigens against background using standard immunohistochemistry methods such as immunofluorescence.

We thank the Reviewer for their close reading of the work, and appreciate their comments to improve this manuscript.

R2.2: Does the work support the conclusions and claims, or is additional evidence needed?

- Yes the work supports the claims and conclusions. Denoising with DIMR and DeepSNF appear to reduce background to a comparable degree or in some cases better than other algorithms used in the literature, but also appears to reduce resolution visually.

1. - Fig 1f: some markers seem to disappear after DeepSNF denoising (ie CD31, CD20). Is this biologically true? Were there other analytes tested in the same tissue to corroborate the results (ie CD19 for B cells?)

The Reviewer appears to be asking why some signal is different between the DIMR-processed and the DeepSNiF process portion of the images in Fig.1f. First, we will describe the results in Fig. 1f: here we have placed DIMR and DeepSNiF-processed portions of images directly next to each other for better visual inspection. The lower left half of each image corresponds to DIMR processing of the lower left image region, while the upper right is DeepSNiF processing of the upper right image region. For side-by-side comparison of DIMR and DeepSNiF-processed images to the identical regions on the raw images, please refer to Supp Fig 17, which shows the full field of view of the raw image for each channel shown in Fig.1F.

Normally, the unspecific staining signal should be lower than true signal. However, because of shot noise, some pixels may seem brighter (Supplementary Note 1.2.1 on Page 4 of the Supplementary Materials). The disappearing signal in the channels highlighted by the Reviewer is likely due to the fact that there is no positive non-noise signal there. Morphologically, the distributed signal in the CD20 channel, for example, is representative of a random noise-field, without actual cell-demarcated signals.

Since there are very few cells positive for CD31 and CD20 in the image chosen for Fig 1f, please refer to the images in Supp Fig 19 and Supp Fig 20 (Pages 40 and 41 of the Supplementary Materials), which include CD31 and CD20 shown by analysis with DIMR and DeepSNiF side-by-side to better demonstrate the effects of DeepSNiF on these markers. The imaging panel does include markers that are expected to have either overlapping or mutually exclusive expression patterns, many of which are shown in Fig 4. For example, CD31 and CD61 both have expected staining patterns on megakaryocytes. However, antibody clones for CD19 were previously tested and did not perform as well on decalcified bone marrow tissue as expected based on lymphoid tissue staining.

We have utilized immunofluorescence (IF) images to verify the denoising results. Please refer to the revised Supplementary Fig. 23 on Page 44 of the Supplementary Materials. In this figure, we have used CD3, CD4, CD61 and CD169 to verify our denoising results. The restored IMC images are closer to the IF images stained by the same antibody, both visually and quantitatively.

R2.3: - Fig 2h is showing that CD20+ gated cells express some amount of MPO. Is this a biological truth or is it an artifact of the image segmentation? Same with CD3+ and CD20+ gated populations having some expression of CD11b

This is an important question. The expression of CD11b, CD15, CD71, CD235a and MPO on CD20+ and CD3+ gated cells are caused by multiple reasons. Firstly, 1) Technical imperfections due to sectioning, staining, and segmentation can cause this issue. Tissue sections are 4-6 microns in thickness, and since the full thickness of the section is ablated in IMC, adjacent cells from above or below within the section will appear as overlapping staining patterns. Since MPO and CD11b are bright markers of the most common cell populations, these are most likely to overlap. Such false segmentation may cause the false positive signal in CD3+ and CD20+ cells. Secondly, 2) noise present from the image formation process can contribute to the issue of marker signal presence. We have analyzed this condition mathematically in Supplementary Note 1.2.2 from Page 4 to 5 of the Supplementary Materials. Briefly, the hot pixel noise and the shot noise are likely to generate false positive or false negative single cell data, and further, the shot noise will affect low intensity regions more due to the characteristics of shot noise.

In this paper, we focus on how denoising algorithms can address the second class of artifacts. The first artifact could be mitigated by better segmentation algorithms and sectioning/staining procedures. This is the reason why even after denoising, there are still false positive existing in CD3+ and CD20+ cells. We have added this clarification from Line 282 to 284 of the article file (redline version).

R2.4: Are there any flaws in the data analysis, interpretation and conclusions? - Do these prohibit publication or require revision?

- With the disclosure that I am not an expert on computer algorithms and bioinformatic analysis, it does seem that the data analysis and interpretation support the conclusion, with the caveats

stated above

Is the methodology sound? Does the work meet the expected standards in your field?

- Yes, the initial segmentation analysis is based on a well-established pipeline published by the Bodenmiller group

Is there enough detail provided in the methods for the work to be reproduced?

- Yes

We thank the Reviewer for the positive assessments. We have also included Supplementary and external (GitHub) data and pipeline information in order to have this work reproduced on these and other data sets.

R.2.5: Additional comments/questions: Authors are encouraged to discuss the questions posed below: 3. - What is the maximum amount of “noise” before a channel is deemed unable to denoise? Is there an empirical way to determine this?

This is an interesting question. In theory, there is no maximum noise level preset for denoising algorithms. Even under some extremely noisy conditions (CD20 and CD3 in Fig. 1f, and most of the markers in Supplementary Figs. 18 and 20 on Supp Pages 39 and 41 respectively), DeepSNiF can still improve the image quality.

Nevertheless, we have noticed and tested our approach on many images and datasets. We have found that the lower the SNR of the raw images, the more information has been lost and thus the quality of the restored images are lower (Supplementary Figs. 6-11 from Page 26 to 31 of the Supplementary Materials). Another aspect of this question or issue is that denoising of very low SNR images will produce images with less detail than that of high SNR input data. Thus, it is possible that even if ‘maximally’ noisy data can be denoised, the images may not be of sufficient quality for analysis. However, even though some of the low SNR IMC imaging data present in this manuscript may have a lower image quality (ie. Blurriness), our experiments throughout this work have validated that such denoised images can still enhance downstream analysis. We have added this discussion on Line 487-491 of the revised manuscript (redline version).

R2.6: - Have analytes that have low signal-to-noise ratio been tested?

Yes, we have tested low SNR images. A major motivation for undertaking of the work in this manuscript is due to the fact that in highly complex tissues like the (healthy and diseased) marrow and many tumor tissues are difficult to assess because of low SNR image data in many channels of interest. We have pursued IMC as a method because other techniques, namely IF, fail in some of these difficult to image tissues in multiplexed workflows. This may be due to the need for repeated wash steps, and the overpowering autofluorescent background. In the presented IMC work, please refer to CD20 and CD3 channels in Fig. 1f and most of the markers in Supplementary Figs. 18 and 20 (Pages 39 and 41 of the Supplementary Materials) for examples of low SNR IMC data that we have processed. We also tested the denoising results under different SNR conditions with simulated data (Supplementary Figs. 6-11 from Page 26 to 31 of the Supplementary Materials).

R2.7: - How would the pipeline adapt to analytes that should visualize targets localized in-between nucleated cells (ie. synapses)? Denoising these channels often results in reduction of image intensity in areas where a signal is expected.

Small areas of staining at the size of a sub-cellular synapse (e.g. 1-2 microns in diameter) will not be successfully distinguished by IMC due to its relatively low resolution of 1 micron, according to Nyquist sampling theory. Therefore, it is likely that a deep learning network cannot learn the features of such small structures. Instead, we and others have tested markers with interstitial staining patterns (e.g. vessels, fibrosis, reticular cytoplasmic projections). As we demonstrate in this work, these can be effectively restored using our hot pixel removal and denoising pipeline (for example: CD31, CD34 and Collagen III in Figs. 1, 2 and Supplementary Figs. 19, 20 (Pages 40 and 41 of the Supplementary Materials)). In particular, the data in Fig. 2b-c have demonstrated that our denoising algorithm achieves higher signal segmentation accuracy than raw images. We have added this discussion on Line 449 to 454 of the article file (redline version).

R2.8: - How would the software handle instances where cell signal is unexpected; ie if a novel cell type is discovered, how would they distinguish between true positive and false positive?

The cell types in multiplexed images are determined by multiple single marker expressions. Therefore, the answer is determined by the data in each single marker channel. Compared to raw inputs, IMC-Denoise robustly improves image quality, removes background signal and maintains true signals across individual single marker images (Fig. 2b,c and Supplementary Figs. 23, 28b, 29b, 30b (Pages 44, 49-51 of the Supplementary Materials)). At the same time, IMC-Denoise improves single cell data while maintaining the original scale and linearity (Supplementary Figs. 31-33 from Page 52 to 54 on the Supplementary Materials). Therefore, IMC-Denoise is capable of assisting in the delineation and study of new cell types by enhancing the quality of each single marker data. We submit that complementary, or orthogonal, methods would be needed to confirm such findings by RNAseq, proteomic analyses or other methods.

R2.9: Has this been tested on tissues where there is a mix of morphologically heterogenous cells? (ie brain tissue where there are a mix of glial cells, immune cells, and fibroblasts).

We have performed several experiments to verify that the algorithm can work on the tissues with morphologically heterogenous cells.

- 1) We trained our network with multiple markers (Supplementary Fig. 45 on Page 66 of the Supplementary Materials). The results demonstrates that DeepSNiF works on the markers stained for morphologically heterogeneous markers if the variant features have been learnt in the training process.*
- 2) Monocytes/macrophages are morphologically heterogeneous so that the successful denoising of CD14/CD169 (Fig. 1f) validates the adaptability of DeepSNiF as well.*

In fact, the networks can learn all the features existing in the training images. Therefore, no matter what the cell shapes are, the network can restore the images if all the essential features existed in the training set. We have added this discussion on Line 445 to 450 of the article file (redline version).

R2.10: What happens when there are processes extending from glial cells that are present in the section but their corresponding nuclei are not in the same plane?

The DeepSNiF training and denoising algorithms are not based on nuclear signal or morphology. Although the glial cells are too rare in the tissue types included here (bone marrow, breast, and pancreatic) tissues, we have successfully implemented this algorithm for other cell types where the nucleus is not present within the tissue section (endothelial cells, macrophage projections) or for fine interstitial structures (collagen III). We found that the

network works on such structures (CD31, CD34 and Collagen III in Figs. 1, 2 and Supplementary Figs. 19, 20 (Pages 40 and 41 of the Supplementary Methods)). We have concluded that the networks are able to learn all the features existing in the training set but not focus on any single specific structures. We have added this discussion on Line 449 to 452 of the article file (redline version).

Reviewer #3:

This study develops IMC-Denoise, a denoising automated pipeline to enhance Imaging Mass Cytometry (IMC) images. Two primary sources of noise are here addressed: hot pixels and shot noise. To remove hot pixels, IMC implements a differential intensity map-based restoration (DIMR); to filter shot noise, it implements a deep learning method, DeepSNF. The approach presented here is of intrinsic interest as it develops some new algorithms and approaches that are illuminating and useful. The extensive benchmarking and comparisons with existing denoising methods are strengths of the study. The end-to-end analysis, including the impact on automated cell phenotyping, is important and further strengthens the study. Some concerns that should be addressed to improve the interpretability and, hopefully, the utility of the study are listed below.

R3.1: How many iterations of DIMR does it take to adequately remove the hot pixels in data? Some quantification of the variability would be very helpful. (The paper indicates $n = 4$ and iteration number is set at 3.) How computationally intensive is the approach?

We thank the reviewer for helping us clarify this point for the readers. To address this question, we have utilized simulated data to test the iteration times and the running time of DIMR algorithm. The results are summarized in the new Supplementary Figure 2 and new Supplementary Note 3.3.1 on Page 22 of the Supplementary Materials.

New Supplementary Fig. 2 DIMR algorithm evaluations on the simulated DNA dataset ($N = 50$ per noise setting). (a) Evaluation of DIMR with Niter from 1 to 5. "Same results" indicate the iteration stops beforehand. For instance, in noise setting 3, DIMR stops after the second iteration even though N_iter is larger than 2. (b) DIMR running time evaluations under different noise settings with $N_iter=3$.

The results indicate that when hot pixel density is high, the accuracy almost does not improve after 3 iterations. In the low hot pixel density condition, the accuracy does not change after a first or second iteration. In fact, the iteration can stop beforehand when no changes are

detected between different iterations (Supplementary Algorithm 1 on Page 9 of Supplementary Materials). To summarize, we recommend $N_{iter} = 3$ in DIMR.

We add that it normally takes 0.05s to 0.4s to implement DIMR for a 500x500 image, depending on the hot pixel density. This indicates that DIMR is not computationally intensive, and can fulfill the requirement of ex vivo implementation on IMC images. We have also revised the corresponding text from Line 134 to 136 of the article file (redline version).

R3.2: The definition of the threshold point x_T (top of page 6) as the value of x at which the second derivative, evaluated at $x-dx$, of the fitted curve from the kernel density estimation algorithm is greater than or equal to 0 (while the second derivative evaluated at x is less than or equal to zero) is mathematically imprecise and problematic. The " $x-dx$ " bit is problematic. Please clarify this definition.

To clarify this definition, we have replaced dx as Δx . Here Δx represents a small value. Because the pixel values of the raw images are discrete, Δx is normally set as 1. We also revised the corresponding text from Line 127 to 129 of the article file (redline version).

R3.3: In the expression for the loss function of the DeepSNP model: The pixel mask is missing from equations 2 (and equation 31) in the regularization term (with coefficient given by the Hessian norm regularizer).

We thank the reviewer for this comment. In fact, the pixel mask only works on the data fidelity term, while the regularization term works on all of the pixels of each patch. This is because we assume all the pixels in IMC images hold this regularization characteristic. To make this more clear, we have revised the corresponding part from Line 161 to 162 of the manuscript (redline version).

R3.4: It would be useful for the authors to provide a justification for the specific choice of neural network architecture. The deep learning methodology is given in very general terms and is somewhat of a blackbox.

This is an important point. We have revised the network description section in Methods, entitled "Neural network implementation". Please refer to Lines 556 to 569 of the manuscript (redline version). We outline the decision making process and provide stepwise context for the selection of architectural components.

R3.5: The manuscript can also benefit from thorough language editing.

We have thoroughly edited the manuscript.

R3.6: Other: In description of equation 2 in the paragraph immediately below, shouldn't the M be replaced by M_p (i.e. the pixel mask). M is not a separate variable in equation 2.

We thank the reviewer for this helpful observation. We have replaced M by M_p . Please refer to Line 159 of the article file (redline version).

Reviewers' Comments:

Reviewer #1:

Remarks to the Author:

I would like to thank the authors for the great effort they have put into their revisions. By adding additional comparisons and figures, they have been able to take away my concerns regarding comparisons and changes to the data. The result is a strong manuscript presenting a well validated method to improve IMC data analysis. The addition of the work flow in Supp Fig 12 and the tutorial on Github is a significant step towards making the tool accessible to other users.

Reviewer #2:

Remarks to the Author:

The authors have provided direct and thorough answers to the questions posed by this reviewer. The authors provide examples and have revised Figures, addressing the concerns raised. I am very satisfied with the revised version of this manuscript and I recommend publication of this important work.

Reviewer #3:

Remarks to the Author:

The authors have addressed my concerns from the previous round.

Please find enclosed our responses to the resubmitted manuscript for our paper, entitled "IMC-Denoise: a content aware denoising pipeline to enhance Imaging Mass Cytometry".

Response to Reviewers:

Reviewer #1 - I would like to thank the authors for the great effort they have put into their revisions. By adding additional comparisons and figures, they have been able to take away my concerns regarding comparisons and changes to the data. The result is a strong manuscript presenting a well validated method to improve IMC data analysis. The addition of the work flow in Supp Fig 12 and the tutorial on Github is a significant step towards making the tool accessible to other users.

Author Response: We thank the Reviewer for their generous comments. We are eager for the field to utilize this technology, and we think that the tutorial will help to encourage users to experiment with our pipeline.

Reviewer #2 - The authors have provided direct and thorough answers to the questions posed by this reviewer. The authors provide examples and have revised Figures, addressing the concerns raised. I am very satisfied with the revised version of this manuscript and I recommend publication of this important work.

Author Response: We thank the Reviewer for their comment.

Reviewer #3 - The authors have addressed my concerns from the previous round.

Author Response: We thank the Reviewer for their comment.